# Rejuvenation as the origin of planar defects in the CrCoNi medium entropy alloy

Yang Yang [1,2] ✉, Sheng Yin [3], Qin Yu [3], Yingxin Zhu[2], Jun Ding [4], Ruopeng Zhang[5], Colin Ophus [1], Mark Asta [3,5], Robert O. Ritchie [3,5] & Andrew M. Minor [1,5] ✉

High or medium- entropy alloys (HEAs/MEAs) are multi-principal element alloys with equal atomic elemental composition, some of which have shown record-breaking mechanical performance. However, the link between short-range order (SRO) and the exceptional mechanical properties of these alloys has remained elusive. The local destruction of SRO by dislocation glide has been predicted to lead to a rejuvenated state with increased entropy and free energy, creating softer zones within the matrix and planar fault boundaries that enhance the ductility, but this has not been verified. Here, we integrate in situ nanomechanical testing with energy-filtered four-dimensional scanning transmission electron microscopy (4D-STEM) and directly observe the rejuvenation during cyclic mechanical loading in single crystal CrCoNi at room temperature. Surprisingly, stacking faults (SFs) and twin boundaries (TBs) are reversible in initial cycles but become irreversible after a thousand cycles, indicating SF energy reduction and rejuvenation. Molecular dynamics (MD) simulation further reveals that the local breakdown of SRO in the MEA triggers these SF reversibility changes. As a result, the deformation features in HEAs/MEAs remain planar and highly localized to the rejuvenated planes, leading to the superior damage tolerance characteristic in this class of alloys.

Plastic deformation in crystalline materials occurs via structural changes known as defects such as dislocations, SFs, and TBs. The reversibility[1–3] of these defects typically depends on the defect formation energy[4,5]. When the defect formation energy is high and positive, the system prefers to remove the defects when the external driving force is released, thus minimizing its total free energy. When the defect formation energy is relatively small positive or even negative, defects tend to maintain their original shape after the release of the external driving force, leading to irreversible changes in materials. The defect formation energy has significant implications for the deformation mechanisms and mechanical performance of materials[4,6].

For traditional alloys, where there are only one or two principal elements with several trace elements, the defect formation energies are usually fixed when the temperature is fixed. Recent studies of HEAs/MEAs[7–11] have expanded this view[12,13], as theoretical studies showed that SRO[14–16] can tune the SF energy[17] in a wide range from negative to positive in the same material, stabilizing heterogeneous structures at the nanoscale with the co-existence of multiple dominating deformation mechanisms depending on the local SF energy. The existence of SRO in HEAs/MEAs lowers the entropy and free energy of the system, and a subsequent increase in SF energy[14] is confirmed by experimental observations[15]. However, a fundamental question remains as to

[1]National Center for Electron Microscopy, Molecular Foundry, Lawrence Berkeley National Laboratory, Berkeley, CA, USA. [2]Department of Engineering Science and Mechanics and Materials Research Institute, The Pennsylvania State University, University Park, PA, USA. [3]Materials Sciences Division, Lawrence Berkeley National Laboratory, Berkeley, CA, USA. [4]Center for Alloy Innovation and Design (CAID), State Key Laboratory for Mechanical Behavior of Materials, Xi'an Jiaotong University, Xi'an, China. [5]Department of Materials Science and Engineering, University of California, Berkeley, CA, USA. ✉e-mail: yangyang@alum.mit.edu; aminor@berkeley.edu

whether and how this interplay of SRO and crystalline defects in HEAs/MEAs leads to their outstanding mechanical properties.

Paradoxically, as the degree of SRO reduces, the system will reach a higher energy state. This state has a lower SF energy which is intrinsically more ductile[13]. In this vein, the destruction of SRO in HEAs/MEAs can be considered a form of rejuvenation (see Supplementary Fig. 1), which has been widely discussed as a beneficial structural transition in disordered materials such as bulk-metallic glasses (BMGs)[18,19]. When BMG is rejuvenated, the structure exhibits a lower degree of ordering and a higher energy state than its relaxed counterparts, resulting in enhanced ductility and toughness. In the CrCoNi MEA system, Li et al. predicted that it only takes three Burgers vectors of dislocation glide to destroy the SRO completely using MD simulations[13]. Therefore, it is expected that the collective movements of dislocations can randomize the atomic arrangement and rejuvenate the system, increasing the propensity of long and stable SFs in HEAs/MEAs due to a lower SF energy. Despite the extensive efforts[14–16,20–22] in understanding SRO in HEAs/MEAs by advanced characterization techniques, there is still a lack of experimental evidence showing how rejuvenation in HEAs/MEAs is tied to the sequence of deformation mechanisms that generate the long, thin planar defects characteristic of HEAs/MEAs.

To address this crucial knowledge gap, it is imperative to in-situ probe the evolution of SFs and TBs in HEAs/MEAs under controlled straining conditions, with a combination of high spatial resolution to resolve individual defects and a wide field of view to enable accurate statistical analysis. Nevertheless, previous techniques[1,3,23] such as high-resolution transmission electron microscopy (TEM), TEM dark-field imaging, or electron channeling contrast imaging (ECCI) encounter various challenges, such as a restricted field of view, a high rate of radiation damage, a low spatial resolution, or the inability to map the strain distribution or distinguish between different types of planar defects.

Here, we have developed a characterization technique, by combining nanomechanical testing[24–26], in situ energy-filtered 4D-STEM[27,28], and advanced defect classification algorithms, as illustrated in Fig. 1. This approach enables us to simultaneously tackle the limitations of previous methods and provide a comprehensive understanding of the evolution of SFs and TBs in metals near a crack tip during deformation at room temperature. Using this technique, we have directly probed the reversible to irreversible transitions of SFs and TBs in an equiatomic CrCoNi MEA due to rejuvenation, revealing critical insights into the role of SRO on the deformation process.

## Results

A water-quenched equiatomic CrCoNi MEA with SRO and a pure Ni single crystal sample used for comparison, were selected for the study. We chose Ni as a reference material because it is a constituent element of the CrCoNi MEA under this study and shares the same face-centered cubic (FCC) crystal structure. Given its high and stable SF energy, Ni provides a stark contrast to CrCoNi MEA, whose SF energy varies widely depending on the degree of SRO. The samples were firstly twin-jet electrochemically polished to be electron transparent. Thin foils with [110] as the surface normal were lifted out by focused ion beam (FIB) and transferred to push-to-pull (PTP) microelectromechanical system (MEMS) chips, as illustrated in the scanning electron microscopy (SEM) images in Fig. 1a, b. Both the MEA and the pure Ni sample transferred to the MEMS chip are single crystals with the same orientation. The thin samples were further patterned by FIB to have smooth fillets at the sides and a sharp crack at the center, as shown in Fig. 1c. This setup enables uniaxial tensile testing of the thin specimen in the TEM. During the FIB transfer and patterning process, we used a special method that prevents ion beam damage and Ga contamination in the region of interest (ROI) in the sample (see Method for more details).

In a 4D-STEM experiment, a nano-sized electron beam rasters across the sample so that at each real space point on the sample an unfiltered nanobeam electron diffraction (NBED) pattern is formed, as illustrated in Fig. 1a. An electron energy filter then selects the electrons with nearly-zero loss of energy to form an energy-filtered NBED pattern, with the signal-to-noise ratio greatly improved due to the removal of inelastically-scattered electrons. The energy filter is critical for capturing weak diffuse signals from SFs, which would otherwise have a low signal-to-noise ratio or even be invisible by traditional imaging methods. All energy-filtered NBED patterns corresponding to each real space point are recorded by a fast direct electron detector for analysis shown in Fig. 1d. After detection and classification of the Bragg peaks, we summed the characteristic Bragg peaks in each NBED pattern to further enhance the signal-to-noise ratio of the diffuse signals from planar defects, allowing us to precisely differentiate three different structures in the sample (Fig. 1d), including matrix, SF and TB. The SF shows characteristic diffuse streaks along the [1–11] direction, while the TB shows a characteristic (111) Bragg peak next to the (002) Bragg peak of the matrix. Figure 1e shows a comparison between the conventional technique, i.e., STEM high angle annular dark-field (HAADF) imaging, with our techniques, based on the characterization of the same region on an equiatomic CrCoNi MEA after 1000 cycles of mechanical deformation. The location of SFs and TBs is nearly invisible in the STEM HAADF image, while it is clearly mapped by energy-filtered 4D-STEM. In addition, energy-filtered 4D-STEM enables the mapping of the relative elastic strain at nanometer resolution.

Our results of the cyclic mechanical loading experiments of CrCoNi MEA and Ni are presented in Figs. 2 and 3. In particular, Fig. 2a–h show the 4D-STEM defect mappings in CrCoNi MEA and Ni at different deformation stages, while Fig. 2i–p delineate the corresponding schematic drawings to highlight the critical nano-structural evolution. Four different stages of deformation, i.e., time ($t$) at 0, T/2, T, and 1000 T, respectively, were analyzed with energy-filtered 4D-STEM, as illustrated in Fig. 2q, where T is the length of time per cycle. Quantitative analysis of these 4D-STEM data is shown in Fig. 3. The strain mappings for the CrCoNi MEA and Ni are shown in Supplementary Figs. 2 and 3, respectively. At $t = 0$, both the inherent as-quenched CrCoNi MEA and Ni showed a small degree of SF content and no twinning ahead of the crack tip. At $t = T/2$, the force is increased to the maximum load in the first cycle. A significant number of SFs have been generated ahead of the crack tip in CrCoNi MEA, with the average SF length increased from 17.5 nm to 48.3 nm (Figs. 2b, j and 3a). In contrast, the SFs in Ni have very limited growth in population and average length (Figs. 2f, n, 3a). The comparison between CrCoNi MEA and Ni at $t = T/2$ indicates that the SF energy ($\gamma_{SF}$) is higher in Ni than in CrCoNi MEA, which agrees with previous experimental measurements[29,30] where $\gamma_{SF}^{Ni}$ and $\gamma_{SF}^{CrCoNi}$ are approximately 120 mJ/m² and 13.3 mJ/m², respectively. At the end of the first cycle ($t = T$), the load was released. The SFs in both the CrCoNi MEA and Ni retracted prominently, recovering to a state similar to $t = 0$ (Figs. 2c, g, k, o, 3a). In the CrCoNi MEA, the average SF length was reduced to 22 nm. The observation of reversible SF activity from $t = 0$ to $t = T$ implies that SFs are not thermodynamically favorable due to the positive SF energy. However, after finishing 1000 cycles of loading and after the external load was set to 0 ($t = 1000$ T), we observed an outstanding difference between CrCoNi MEA and Ni. Profuse SFs and twins manifest in the MEA, with the average lengths of SF and twins reaching 67.1 nm and 124.2 nm, respectively (Figs. 2d, h, l, p, 3a). The significant increase of SF length and population in the CrCoNi MEA even at zero external load at $t = 1000$ T compared to $t = T$ indicates that the SFs become more energetically favorable to form. According to our previous discussion, this should be related to a major reduction in SF energy, suggesting that the region has undergone a rejuvenation, i.e., it is more like a random solid solution. These time-dependent dynamical changes in the SF energy in CrCoNi MEA are distinctly different from the

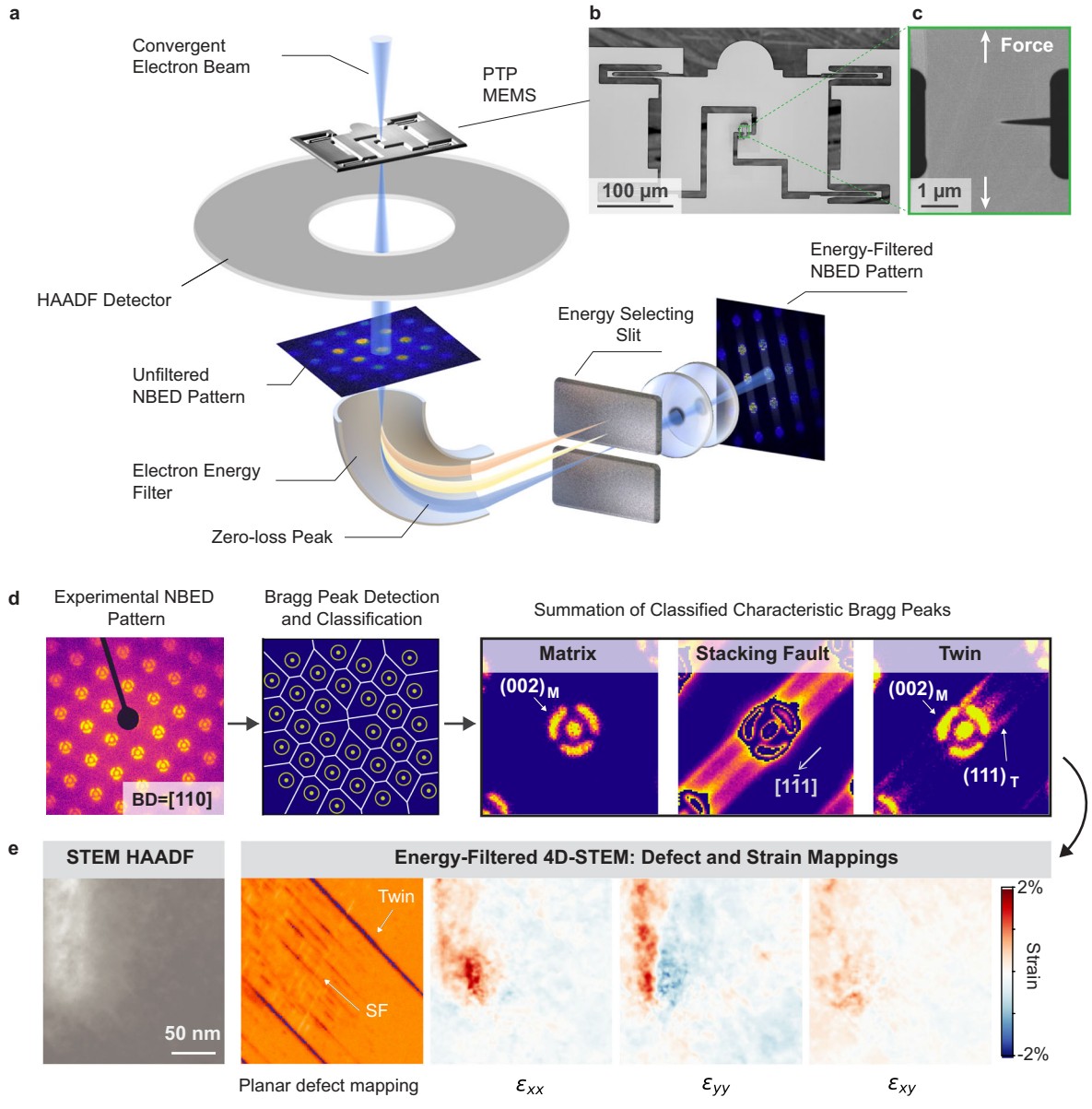

**Fig. 1 | In-situ energy-filtered four-dimensional scanning transmission electron microscopy (4D-STEM) coupled with nanomechanical testing to directly visualize the evolution of stacking faults (SFs) and twin boundaries (TBs) during mechanical deformation. a** Schematic illustration of the experimental setup. **b** Scanning electron microscopy (SEM) image showing the push-to-pull (PTP) device for cyclic tensile testing. **c** SEM image showing an equiatomic CrCoNi medium entropy alloy (MEA) sample with a notch before mechanical testing. The loading direction is indicated by the white arrows. **d** Illustration of the routine to process the 4D-STEM data, which classified the nanobeam electron diffraction (NBED) patterns into 3 categories with unique features: matrix, SF and TB, respectively. **e** Comparison of microscopy images acquired on the same region in a CrCoNi MEA sample after 1000 cycles of deformation. The edge of the crack notch is at the top-left corner of each image. Energy-filtered 4D-STEM enables the precise mapping of planar defects and strain at nanometer resolution, which was not available by traditional imaging techniques such as high-angle annular dark-field (HAADF) STEM.

observations in Ni, as Ni barely showed any increase in average SF length at $t = 1000$ T.

To understand the impact of crack tip rejuvenation on crack propagation, we have plotted the evolution of the normalized crack-tip opening displacement (CTOD) in the CrCoNi MEA and Ni in Fig. 3b. After the first loading cycle, the amount of crack blunting is similar for CrCoNi MEA and Ni. However, after 1000 cycles, the crack-tip blunting is more significant in the CrCoNi MEA than Ni, suggesting that rejuvenation promotes the ductility near the crack and facilitates crack blunting to reduce the stress concentration. This agrees with our observation of longer SFs in a rejuvenated sample (Figs. 2d and 3a). When the length of SFs is greater, the spacing between partial dislocations increases, leading to a larger dissociation width. This results in a higher density of dislocation interactions and a more complex dislocation network, which in turn enhances work hardening. The increased work hardening rate contributes to improved strength, ductility, and toughness in the material. We further studied the evolution of the average hydrostatic strain in a squared region ahead of the crack tip (Fig. 3c). It was found that the residual hydrostatic tensile strain ahead of the crack tip kept increasing in Ni during cyclic loading, while that in the CrCoNi MEA shows an opposite trend. In the CrCoNi MEA, the rejuvenation activated the SF-mediated plasticity, effectively releasing the detrimental hydrostatic tensile strain localized near the crack tip and thus reducing the propensity for crack propagation. In contrast, Ni relies on full-dislocation mediated plasticity, providing less effective stress relief ahead of the crack.

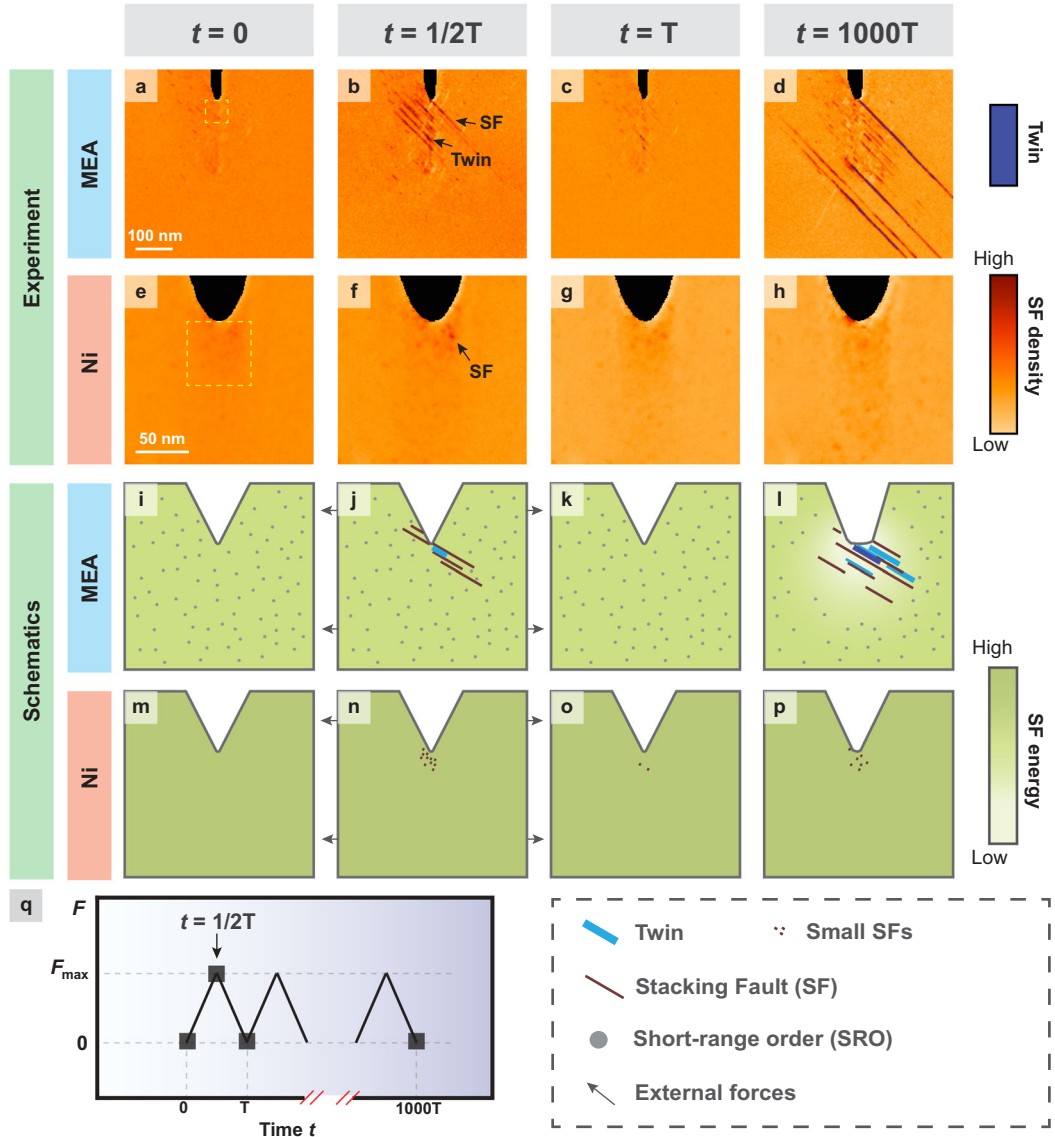

**Fig. 2 | The evolution of SFs and TBs in CrCoNi MEA and pure Ni during cyclic loading up to 1000 cycles. The definition of time ($t$) is illustrated in (q).** **a–h** Comparison of the 4D-STEM defect mappings in CrCoNi MEA and pure Ni at different stages of deformation. TBs are highlighted in blue color, while the relative SF density is indicated by the orange color bar. **i–p** Schematic illustration of the nanostructural evolution in the CrCoNi MEA and Ni during cyclic loading. **q** The cyclic mechanical loading curve for our experiments, where four characteristic points were selected for 4D-STEM characterization. T is the length of time per cycle.

The rejuvenation of CrCoNi MEA after cyclic loading can be explained by the destruction of SRO due to the dislocation-SRO interactions. To provide insight into this mechanism, MD simulations were performed. First, simulations of single-cycle loading-unloading were conducted to compare CrCoNi MEA with and without SRO, as illustrated in Supplementary Fig. 4 and Supplementary Movies 1, 2. The findings provide compelling evidence that the reversibility of SFs is linked to the presence of SRO in CrCoNi MEA; in its absence, SFs tend to be irreversible. Secondly, cyclic loading tests were simulated to explore the interactions between dislocations and SRO. The snapshots in Fig. 4a–c show the evolution of deformation in front of the crack tip, including dislocation nucleation, SFs formation, and twinning. The FCC and HCP phases are colored green and magenta, respectively. The formation of SFs is highly reversible in the first 5-7 cycles, indicating a positive SF energy, as shown in Fig. 4 and Supplementary Movie 3. During the cyclic deformation, dislocation activity and SFs formation occur close to the crack tip region, leading to a decrease in SRO as the cycle number increases, as shown in Fig. 4d. The reduction of the degree of SRO due to deformation leads to the continuous reduction

of general SF energy near the crack tip. This continuous reduction of SRO during cyclic loading also suggests that the reversible SF extension and retraction are pseudoelastic by nature, as the glide of trailing partials and leading partials will inevitably distort the SRO arrangement locally, which may not be fully recovered to its initial state during the unloading process. After around 20 cycles in the simulations, the SF length started to increase dramatically, indicating that the SF energy has dropped to a threshold value, below which SFs become a dominating plasticity mechanism in CrCoNi MEA. To gain a deeper understanding of which type of SRO is more susceptible to damage by cyclic loading, we conducted a study on the evolution of the Warren-Cowley parameters[31]. These parameters are widely utilized to describe preferred or unfavored atomic pairings within a specific nearest-neighbor shell. Positive values of the Warren-Cowley parameter indicate an unfavored tendency of atomic pairs, whereas negative values suggest a favored tendency. A zero value implies the absence of SRO. Our analysis of the first nearest-neighbor shell Warren-Cowley parameters, as illustrated in Fig. 4e, reveals that the Ni-Cr and Co-Co pairs are insensitive to mechanical loading. Conversely, the other types of

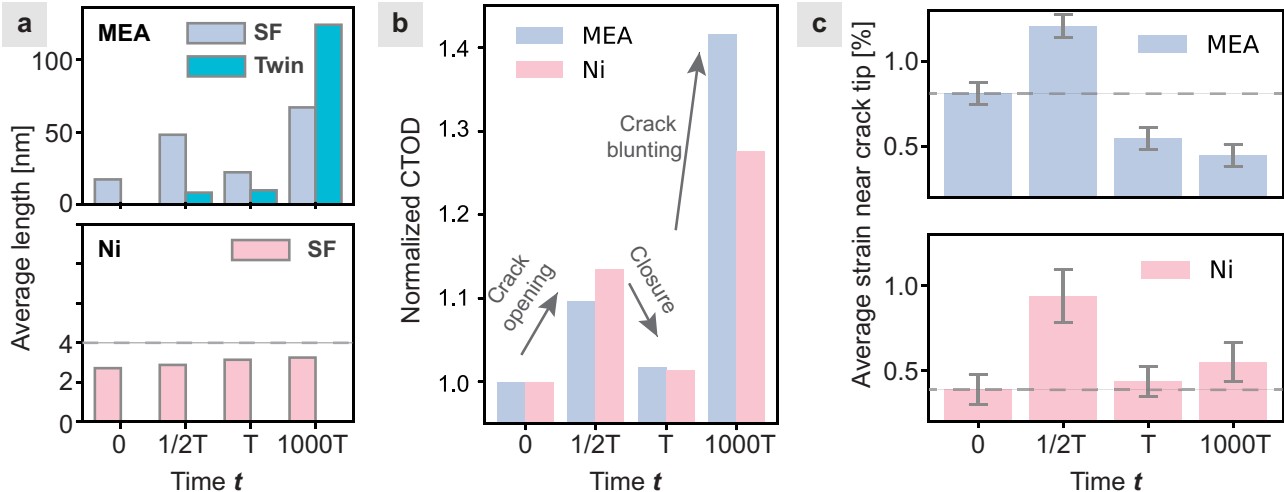

**Fig. 3 | Quantitative analysis of the rejuvenation in CrCoNi MEA during the cyclic mechanical deformation. a** The evolution of the average planar defect length at different stages of deformation in CrCoNi MEA and Ni. **b** Comparisons of the normalized crack-tip opening displacement (CTOD) in the CrCoNi MEA and pure Ni. Both the cracks in the CrCoNi MEA and Ni become significantly blunter after 1000 cycles. **c** The evolution of the average hydrostatic strain in a squared region ahead of the crack tip. Tensile strain shows a positive value. The side length of these regions is 60 nm. These two regions are also highlighted by the yellow boxes in Figs. 2a and 2e. The error bar indicates the standard deviation. T is the length of time per cycle.

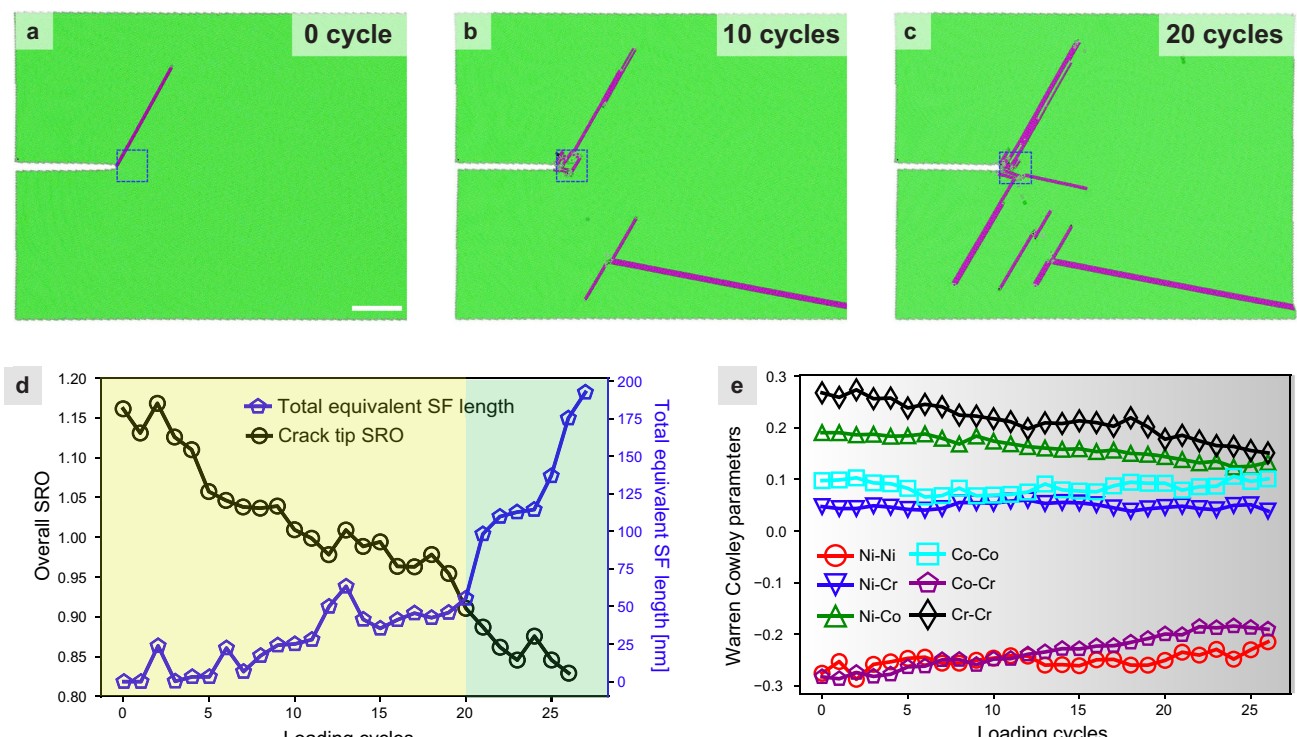

**Fig. 4 | MD simulation to reveal the evolution of SRO and SFs in an equiatomic CrCoNi MEA during the cyclic loading process. a–c** Snapshots showing the evolution of deformation in front of the crack tip, including dislocation nucleation, SFs formation, and twinning. The FCC and HCP phases are colored green and magenta, respectively. Scale bar, 10 nm. **d** The evolution of SRO and the total equivalent SF length in front of the crack tip region (a 5 nm × 5 nm zone as shown by the blue dashed box in **a–c**) during the cyclic process. **e** The evolution of the first nearest-neighbor shell Warren Cowley parameter in the boxed zone in **a–c** during cyclic loading.

SRO exhibit a steady decrease as the cycle number increases. This finding suggests that mechanical deformation may lead to a variation in SRO different from what is typically observed following annealing.

Last but not least, the SF evolution during cyclic loading in MEA revealed by our MD simulation is compared with a previous MD simulation[32] of cyclic loading in pure Cu, which has a lower SF energy

(around 55 mJ/m²)[33] than pure Ni. We analyzed the snapshots of MD simulations in ref. 32 and qualitatively plotted the variation of the estimated total SF length as a function of loading cycles for pure Cu. The results have been included in Supplementary Fig. 5, which compares pure Cu with CrCoNi MEA. In pure Cu, the total SF length consistently increases rapidly with each loading cycle, except for the last

data point, where crack propagation and void formation become dominant (Supplementary Fig. 5a). By contrast, CrCoNi MEA demonstrates a two-phase evolution characterized by a slow increment in SF length initially, which then accelerates, correlating with the progressive breakdown of SRO and rejuvenation in the material.

## Discussion

The transition from reversible to irreversible SF behavior during cyclic loading is pivotal, and its mechanism is essential for understanding the SRO-dislocation interactions that govern the rejuvenation process. The transition can be interpreted from two perspectives.

Firstly, from a free energy standpoint, the SF energy is a critical determinant of SF stability—the lower the SF energy, the more stable the SF. The reversible to irreversible transitions suggest a corresponding shift from higher to lower SF energy. In this work, we first postulate that this shift results primarily from the dislocation glide destructing the SRO, thereby progressively reducing the SF energy and softening the glide planes. This has then been verified by our MD simulations.

Secondly, we may benefit from the view of SF length. An SF in FCC materials is the area between two Shockley partial dislocations. The length of the SF depends on how far the leading partial can travel. If the SF energy is positive, it induces an attractive force between the two partial dislocations. This force is typically counterbalanced by the repulsive elastic interaction between the partials, in the absence of external forces. When an external force is applied, it disrupts this balance, leading to an increase in the separation distance between the partials. Upon release of the external force, a reverse force occurs because the attractive force resulting from positive SF energy substantially outweighs the repulsive force due to the strain energy of the partials. This reverse force narrows the distance between the partials, diminishing gradually as the separation decreases (thereby reducing SF length). In a purely elastic scenario, the SF length will eventually recover to its initial state after unloading. In HEAs/MEAs, however, the scenario differs due to the presence of SRO, which may be destructed by dislocation gliding—an inelastic process. The reduction in SRO by the leading partial alters the SF energy and therefore the reverse force upon unloading is also altered, leading to a rejuvenated state that provides an asymmetric loading and unloading with regards to the position of the leading partial. Upon retraction, there is an energetic balance between the partial dislocation restoring the SRO at the expense of then increasing the SF energy. Further studies of a pure metal with a low SF energy such as Cu would be interesting for comparison in this respect.

It should be noted that during cyclic loading, dislocations predominantly operate on specific glide planes. As these dislocations glide on their activated glide planes, they invariably disrupt the SRO regions located on those planes. This action can fragment a larger SRO region, leading to a decrease in the average SRO region size but simultaneously increasing the local density of these smaller regions. Moreover, certain regions of SRO, primarily the smaller ones, which do not intersect with an activated glide plane, may remain undisturbed post-loading (Supplementary Fig. 6).

In addition, it is worthwhile to comment on the role of excess vacancies in defect evolution during cyclic loading. Vacancies emitted during cyclic loading can facilitate diffusion and, consequently, increase the degree of SRO. However, as detailed in Supplementary Note 1, the role of vacancy-mediated SRO formation in our experiments is considered minimal and possibly negligible. Indeed, our observations indicate a continuous reduction in SF energy, suggesting a diminishing degree of SRO. This trend points to the predominance of a dislocation-induced SRO destruction mechanism in the deformation process.

The SRO parameters can be influenced by alloy composition and thermal-mechanical processing[34–37]. While this study focuses on an equiatomic CrCoNi MEA, the fundamental mechanisms we have identified—namely, the dislocation-mediated destruction of SRO and its consequent effects on SF energy—may exhibit similar trends in non-equiatomic MEAs/HEAs.

The observation of rejuvenation in a quenched CrCoNi MEA after cyclic destruction of SRO offers critical insights into the impact of SRO on the mechanical performance of HEAs/MEAs, giving us a greater understanding of why certain HEAs/MEAs shows a combination of high strength and ductility. Previous MD simulations have shown that SRO in the CrCoNi MEA can lead to an increase in yield strength[13,38]. Furthermore, recent studies revealed that SRO can promote work hardening and thus enhance ductility[15]. Our observations here suggest that the spontaneous local rejuvenation and softening due to the local breakdown of SRO have a dramatic effect on the local structure that is beneficial for fracture toughness. The local rejuvenation in MEAs during deformation converts the material into a composite material with effectively "soft" and "hard" zones that increase the diversity of plasticity mechanisms (Fig. 5). In the rejuvenated regions, the generation of long, thin SFs and twins leads to a high density of interfaces that enhance strain hardenability. The formation and destruction of SRO serve as the switch between reversibility and irreversibility. Therefore, the presence of SRO in HEAs/MEAs can benefit the overall mechanical damage tolerance of these materials.

Our current study primarily focuses on quasi 2-dimensional (2D) models due to their typical use in in-situ TEM experiments, which only accommodate thin samples, and the more manageable computational load for modeling. In a 3D system, more slip planes are activated, potentially increasing the rejuvenation efficiency. In our quasi-2D study, significant rejuvenation was observed at around 20 cycles of loading, after which SFs became irreversible with a high average length (Fig. 4d). In a true 3D scenario, fewer cycles may be required under similar loading conditions, and it is plausible that even a single loading could lead to the rejuvenation of MEAs/HEAs and alter deformation mechanisms. For instance, in CrCoNi MEA samples damaged by shock loading[39], it was found that profuse SFs/twins form a 3D network, characteristic of rejuvenated MEA with low SF energy. This suggests significant implications of rejuvenation for single loadings in a 3D system.

It is crucial to point out that the superior combination of ductility and strength in CrCoNi MEA is not solely due to low SF energy but also the interplay of soft (locally rejuvenated) and hard zones (with high SF energy) within the material. In a single loading in a 3D scenario, while more glide planes may be softened (rejuvenated) than in the quasi-2D scenario, a significant volume remains unrejuvenated, particularly where dislocations are pinned or locked. Thus, the concept of composite materials comprising "soft" and "hard" zones remains valid for the 3D scenario and single loading.

In summary, through the integration of in-situ energy-filtered 4D-STEM, nanomechanical testing, and MD modeling, our study reveals and elucidates the reversible to irreversible transitions of SF dynamics in CrCoNi MEA during cyclic deformation. This phenomenon is primarily attributed to the destruction of SRO along the glide planes, leading to localized material rejuvenation and the formation of a composite system. This system comprises 'soft' zones with diminished SRO and 'hard' zones where SRO is maintained. Intriguingly, it is the SRO that orchestrates the sequence of deformation mechanisms, fostering a synergy of multiple mechanisms and consequently enhancing the mechanical performance of CrCoNi MEA. Identifying strategies to optimize the degree and distribution of SRO in MEAs/HEAs could be pivotal in coordinating local rejuvenation processes, thereby enhancing their mechanical performance.

## Methods

### Materials and sample preparation

An equiatomic CrCoNi MEA raw ingot underwent argon-arc double melting, after which it was divided into smaller specimens. These

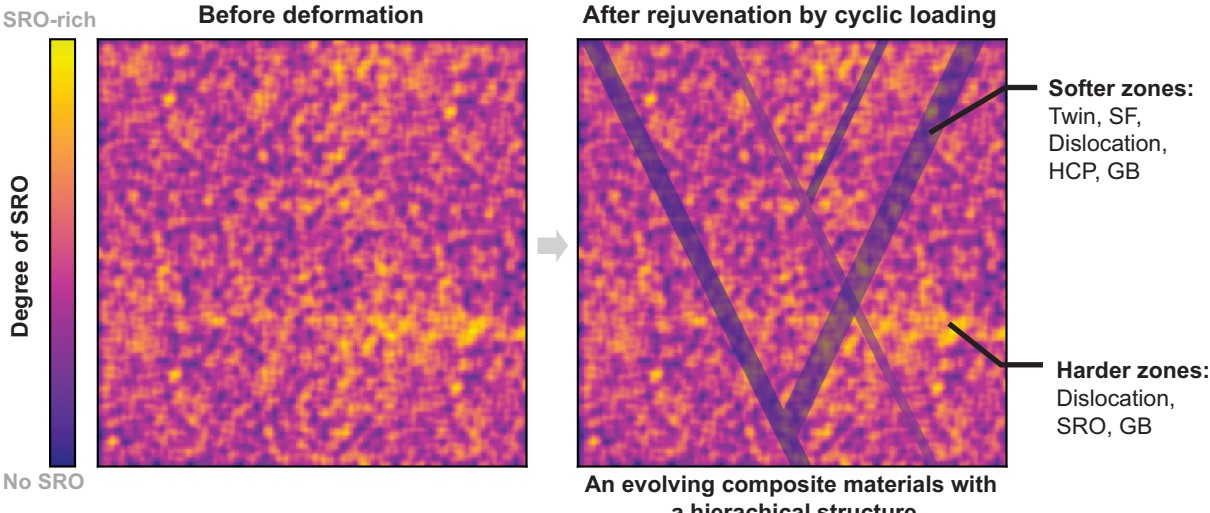

**Fig. 5 | Schematic drawing showing the nanostructural origin of the superior damage tolerance in MEAs and HEAs during the deformation process.** Cyclic deformation gives rise to the emergence of localized rejuvenated regions, which possess a softer nature compared to the surrounding matrix. Consequently, this creates an active composite material characterized by the dynamic interplay of softer and harder zones, which evolve in accordance with the local destruction or formation of SRO. This hierarchical architecture promotes a synergistic activation of diverse deformation mechanisms, facilitating uniform deformation and, ultimately, achieving a balance between high strength and ductility.

specimens were homogenized at a temperature of 1200 °C for 48 h before being water-quenched to room temperature, resulting in a uniform texture and an average grain size of approximately 800 μm, as identified by electron backscatter diffraction (EBSD). X-ray diffraction and EBSD analyses confirmed the presence of a single-phase FCC structure. The Ni single crystal with [110] surface normal was purchased from MTI Corporation.

The bulk MEA and Ni specimens were sectioned and mechanically polished into disks with a diameter of 3 mm. TEM samples were prepared via a two-step process. First, the samples were electrochemically polished using a Fischione twin-jet electropolisher, employing a solution consisting of 70% methanol, 20% glycerol, and 10% perchloric acid at −20 °C. EBSD was performed to identify a large region with the [110] direction of FCC as the surface normal on the twin-jet electrochemical-polished samples. Thermo Fisher Scientific (TFS) Helios G4 FIB was used to transfer a thin and flat foil from the selected grain on the electrochemical-polished sample to a Bruker MEMS PTP device. Then the sample was nanopatterned into the shape shown in Fig. 1 by 30 keV Ga$^+$ beam milling in the FIB. During the FIB transfer and the nano-patterning process, 5 keV electron beam was used for imaging while the ion beam was only used for milling and cutting, minimizing ion beam damage.

**Electron microscopy characterization**
The TFS TEAM-1 microscope, situated at the National Center for Electron Microscopy within the Lawrence Berkeley National Laboratory, was employed for STEM-HAADF and 4D-STEM data acquisition. This double-aberration-corrected microscope was operated at 300 keV. During 4D-STEM experiments, an electron beam of approximately 1–2 nm in diameter scanned the sample, capturing a NBED pattern at each real-space electron beam position. A Gatan K3 direct electron detector, accompanied by a continuum energy filter, facilitated a high signal-to-noise ratio and high-speed data collection (>1000 frames/s) for 4D-STEM data. A specialized bullseye-shaped condenser-2 (C2) aperture was utilized to reshape the electron beam, improving the precision of lattice spacing measurements[40]. The energy filter and bullseye aperture were instrumental in ensuring accurate measurements, as demonstrated in previous research[28]. 4D-STEM experiments were conducted at spot 5 using a 10 μm C2 bullseye

aperture, with a 2 mrad convergence angle and a 105 cm camera length. Upon inserting the bullseye aperture, the monochromator lens setting was adjusted, and the sample ROI was located. The probe current was maximized to enhance the brightness and clarity of higher-order Bragg peaks. Consequently, operation occurred within the 50–100 pA range. Although the probe current is comparable to that of high-resolution STEM (HRSTEM), the dose rate ($e^-$/Å$^2$/s) is approximately 500 times smaller due to the probe's larger size (~1–2 nm). A 15 eV and 20 eV slit were employed for the energy filter in the MEA and Ni experiments, respectively. Scans with step sizes of 1 nm, 2 nm, and 3 nm were conducted to encompass varying fields of view. Additionally, a vacuum scan was executed each day of the 4D-STEM experiments, generating templates for Bragg disk detection. The py4DSTEM package[41] facilitated the analysis of the 4D-STEM dataset. Bragg disk locations were identified by matching the vacuum template to the diffraction disks, enabling the analysis of lattice spacing and strain. The strain zero lattice spacing reference was acquired by averaging the lattice parameter across the entire scanned region before deformation ($t = 0$). The average hydrostatic strain was calculated by the mean of $\varepsilon_{xx}$ and $\varepsilon_{yy}$. The individual evolution of the average of $\varepsilon_{xx}$, $\varepsilon_{xy}$, and $\varepsilon_{yy}$ are provided in Supplementary Figs. 7 and 8. The classification of matrix, twin and SF is achieved by leveraging the unique features in the NBED patterns to generate defect-sensitive virtual dark-field images. A TFS Strata DB235 SEM at 20 kV equipped with an Orientation Imaging Microscopy (OIM) system (Ametek EDAX, Mahwah, NJ, USA) was used for the EBSD characterization. The Bruker Hysitron PI-95 nano-mechanical holder was used to perform the cyclic loading in the TEM. The MEMS PTP chip has a medium stiffness.

**Local chemical short-range order parameter**
For the SRO, we adopted the pairwise multicomponent Warren Cowley order parameter[31], $\alpha_{ij} = 1 - \frac{P_{j,i}}{c_j}$, to quantify the SRO in each specific nearest-neighboring shell. $P_{j,i}$ is the fraction of species $j$ in the nearest-neighboring shell around $i$, and $c_j$ is the concentration of $j$. To indicate the overall degree of SRO, we make use of a quantity given as the sum of all the $\left| \alpha_{ij} \right|$ for all species at nearest-neighbor shell ($CSRO = \sum_{i,j} |\alpha_{ij}|$). Note that the Warren Cowley order parameter[31] we employed differs from the Warren Cowley order parameter used by Li et al.[13], which involves the use of the Kronecker delta function.

## Molecular dynamics simulation

MD simulations were performed using the software package LAMMPS[42] and the atomic configurations were displayed by OVITO[43]. The Embedded Atom Model (EAM) potential for equiatomic CrCoNi was used to describe the interatomic interactions[13]. The dimension of simulation cells illustrated in Fig. 4 is around 800 Å in $x$, 600 Å in $y$ and 50 Å in $z$. The Monte Carlo MD (MC/MD) simulations were first conducted to equilibrate the SRO in the sample before the deformation was applied. During the MC/MD simulation, the periodic boundary conditions were set for all three directions. The samples were initially relaxed and equilibrated at 800 K and zero pressure under the constant-temperature, constant-pressure (NPT) ensemble through MD. After that, MC steps consisting of attempted atom swaps were conducted at the same temperature, hybrid with the MD. In each MC step, a swap of one random atom with another random atom of a different type was conducted based on the Metropolis algorithm in the canonical ensemble. $1 \times 10^2$ MC swaps were conducted at every $1 \times 10^3$ MD steps with a time step of 0.001 ps during the simulation. After the SRO converged in the sample, a 16 nm initial crack was created in the sample before loading by removing two layers of atoms. During the deformation step, only the $z$ direction is periodic and the other two directions are set at a fixed size. The bottom and top two layers of atoms in the $y$ direction were fixed and the cyclic strain was applied to the whole simulation cell. The deformation simulation was conducted under 300 K with a constant-temperature, strain rate of $1 \times 10^8$ s$^{-1}$ and constant-volume (NVT) ensemble applied to all the unfixed atoms.

## Data availability

The data that supports the findings of this study are available from the corresponding author upon request.

## Code availability

py4DSTEM is an open-source package available on GitHub: https://github.com/py4dstem/py4DSTEM.

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

## Acknowledgements

This work was primarily supported by the U.S. Department of Energy, Office of Science, Basic Energy Sciences, Materials Sciences and Engineering Division, through the Damage-Tolerance in Structural Materials program (KC13) at the Lawrence Berkeley National Laboratory (LBNL) under contract No. DE-AC02-CH11231. Work at the Molecular Foundry was supported by the Office of Science, Office of Basic Energy Sciences, of the U.S. Department of Energy under Contract No. DE-AC02-05CH11231. The simulation work made use of resources of the National Energy Research Scientific Computing Center (NERSC), a U.S. Department of Energy Office of Science User Facility located at LBNL, operated under the same contract number, using NERSC award BES-ERCAP0027535. Y.Y. would like to thank Dr. Shaolou Wei from Max-Planck-Institut für Eisenforschung GmbH, Yongwen Sun from the Pennsylvania State University, Dr. Jim Ciston and Dr. Benjamin H Savitzky from LBNL for helpful discussions.

## Author contributions

Y.Y. and A.M.M. conceived the project. A.M.M., M.A. and R.O.R. provided critical guidance on the project. Y.Y. performed electrochemical twin-jet polishing of Ni, small-scale tensile sample preparation and electron microscopy characterization. Y.Y. and Y.Z. performed the analysis of the 4D-STEM data. S.Y. performed the simulations. Q.Y. performed the EBSD analysis and provided critical guidance on the writing of the manuscript. R. Z. performed the electrochemical twin-jet polishing of the MEA sample. Y.Y., S.Y., Q.Y., and A.M.M. wrote the manuscript. J. D. and C.O. provide helpful suggestions on the analysis and writing of the manuscript. All authors contributed to the discussion of the results.

## Competing interests

The authors declare no competing interests.
