## [Peer Review File · Nature Communications]

Rejuvenation as the origin of planar defects in the CrCoNi medium entropy alloyREVIEWER COMMENTS

Reviewer #1 (Remarks to the Author):

This article reports rejuvenation during cyclic mechanical loading in CrCoNi alloy, showing creative softer zones within the matrix and planar fault boundaries that enhance the ductility. Using in situ nanomechanical testing along with 4D-STEM, established by authors, revealed SFs and mechanical twins were reversible, but later they became irreversible. This article showed new perspectives in the field of HEAs/MEAs. However, the critical issues below are also included, especially direct evidence of SRO formation and of SRO destruction are still missing. I, therefore, cannot accept this article publication in this journal.

(1) Major comments

- (i) Direct evidence of SRO formation in the nanomechanically tested samples and of SRO destruction in there should be provided for supporting the authors' hypothesis.
- (ii) Is there any experimental observation to determine SRO type and size?
- (iii) Even though SRO can be destructed by dislocations, small-sized SRO after cutting by dislocations can remain in the system. Hence, the evolution in the density and size of SRO with and without deformation should be clarified.
- (iv) The MD simulation strain rate (typically, $10^7 \sim 10^9 \text{ s}^{-1}$) is several order of magnitude higher than the experimental one. In that case, how to prove that the simulation predictions are able to support the experimental observations?

(2) Minor comments

- (i) Why did the authors select the pure Ni single crystal for comparison, but not select polycrystal Ni?
- (ii) When the temperature is high enough, diffusion to defect formation energy can be reduced. Hence, on page 2 or page 3, this reviewer cannot agree that the defect formation energies are constant.
- (iii) The microstructural discrepancies between the experimental sample and the simulation model. The experimental CrCoNi sample is polycrystalline, while the simulated sample is single crystalline. A bi-/poly-crystal model with grain boundaries is highly recommended to add for comparison.
- (iv) On page 6, what is the meaning of degradation of ordering? Is it the destruction of SRO? Or, reduced SRO degree? It is hard for general readers to understand.
- (v) There are many typos, such as full name of MD in the main text on page 3 and evolution of SFs TBs on page 3. These errors are careless.

Reviewer #2 (Remarks to the Author):

This manuscript addresses an important role of short-range order (SRO) in Medium Entropy Alloys (MEA) during the deformation process by investigating the significant increase of stacking fault (SF) length after thousands of loading cycles via four dimensional scanning transmission electron microscopy (4D-STEM). In reviewer opinion, the manuscript needs a major revision before it could be considered again for Nature Communications. The authors may wish to consider the following points.

(i) The connection between experimental observation with molecular dynamic modelling support needs a clarification. for example, can the EAM potential provides any data for the SF energy of 13.3 mJ/m^2 for CrCoNi MEAs?

(ii) It is also not clear for general readers at which temperature and for which composition of MEAs the authors performed in their experimental conditions. The SRO parameters are very sensitive to both alloy composition as well as finction of temperature and I am wondering whether the main conclusion of this work could be applied for the case where the MEAs composition is not equiatomic (although the authors didn't mention, I assume that they were looking at equal atomic composition). Note also that the MD simulations were performed at 800K where the radom solid solution for this alloys may be

dominant rather than SRO configurations as shown in Fig. b from Ref.

(iii) The authors showed in Figure 4d the destruction of SRO after cycle loading in term of so-called overall SRO deoendence. However, by looking at the different SRO pair dependence in Fig. 4e, it is surprise for me to see the presence of negative SRO trend for Ni-Ni pairs that is contradicting to the positive SRO value of Ni-Ni from Fig. 1b in Ref. 11. Note from the same reference the SRO of Cr-Cr are slightly negative whereas in the present study the authors showed that the are strongly positive.

Reviewer #3 (Remarks to the Author):

The paper illustrates the mechanical rejuvenation of CrCoNi MEA induced by cyclic loading through an in-situ nanomechanical testing method, along with observations using the newly developed energy-filtered 4D STEM technique. The authors directly observe the evolution of defects, such as stacking faults and twins, during cyclic loading. This observation suggests a reduction in stacking fault energy due to the rejuvenation process, which could stem from the destruction of short-range order (SRO) caused by dislocation activities in the crack tip region. The innovative experimental method carries significant potential and could broadly impact nanomechanics studies, contributing to a more profound understanding of materials' fracture and deformation phenomena. However before considering publication, the following points can be addressed.

The mechanism of the transition from reversible to irreversible behavior at certain cyclic loadings should be presented and discussed more clearly. This mechanism is the foundation for the destruction of SRO and the formation of fine, soft (rejuvenated) and hard (chemically ordered) zones at the crack tip. Exploring the transition in relation and the evolution of stacking fault length to cyclic loading frequency and strain amplitude could be intriguing. Also, addressing the possibility of vacancy and interstitial generation during cyclic loading and its impact on defect evolution and stacking fault length would be beneficial.

Considering that the stacking fault energy of Ni is significantly higher compared to MEA, even with SRO, a fair comparison of planar defect evolution between Ni and MEA might be challenging. Authors could consider comparing with Cu for a more comprehensive comparison.

The current demonstrations, both experimentally and in MD simulations, seem to be quasi 2-dimensional. Exploring a 3-dimensional scenario, such as a penny-shaped crack in a 3-dimensional sample, could lead to more complex dislocation activities. Therefore, discussing the effectiveness of single loading for rejuvenation in a 3-dimensional case could provide valuable insights.

An itemized list of responses to reviewers' remarks
(Blue: Reviewer's comments; Black: Our response)

Table of Contents

Reviewer: 1.....2
Reviewer: 2.....15
Reviewer: 3.....22
Reference for point-by-point reply:.....34

(Blue: Reviewer's comments; Black: Our response)

Reviewer: 1

Comment 1:

This article reports rejuvenation during cyclic mechanical loading in CrCoNi alloy, showing creative softer zones within the matrix and planar fault boundaries that enhance the ductility. Using in situ nanomechanical testing along with 4D-STEM, established by authors, revealed SFs and mechanical twins were reversible, but later they became irreversible. This article showed new perspectives in the field of HEAs/MEAs. However, the critical issues below are also included, especially direct evidence of SRO formation and of SRO destruction are still missing. I, therefore, cannot accept this article publication in this journal.

Response.

Thank you for your thorough review and insightful comments. We are grateful for your acknowledgment of the "new perspectives in the field of HEAs/MEAs" that our work introduces. Recognizing the importance of the points you raised, we are happy to address these concerns and enhance our manuscript accordingly.

Comment 2:

Direct evidence of SRO formation in the nanomechanically tested samples and of SRO destruction in there should be provided for supporting the authors' hypothesis.

Response.

Thank you for highlighting the need for direct evidence of SRO formation and destruction in the tested samples. We fully acknowledge the significance of such direct observations.

Direct imaging of CSRO is a grand challenge and may not even be possible in this CrCoNi, as the atomic number and sizes of Cr, Co, and Ni are too close.

Previous theoretical underpinnings have suggested that SRO is a distinct feature of HEAs/MEAs when compared to traditional alloys. Yet, the experimental characterization of SRO in MEAs, particularly in relation to deformation, remains elusive due to the complexities of characterization. Recent publications in journals such as *Nature* have attempted to provide direct evidence of SRO in MEAs through methods like diffuse diffraction signals found in selected area electron diffraction (SAED) patterns^{1,2}. For example, the streaks along {111} direction on the [110] zone axis¹, and the Bragg spots on the $\frac{1}{2}\{-311\}$ location on the [112] zone axis² have been indicated as evidence of SRO in MEAs (See Fig. R1).

Figure R1. Recent research that considered the diffuses streaks or Bragg spots in SAED patterns as evidence of the SRO. Figures reproduced from Ref^{1,2}.

However, these methods have been met with contention. Specifically, a recent comment in *Nature Materials*³ questions the origin of the diffuse signals and suggests they may arise from planar defects rather than SRO (See Fig. R2). In addition, a recent research article in *Nature* considered the diffuse signals as an outcome of reflections from higher-order Laue zones (HOLZs) rather than SRO⁴.

Zone axis	Extra reflections
[011]	Streaking
$[\bar{1}11]$	$\frac{1}{3}\{422\}$
$[\bar{1}12]$	$\frac{1}{2}\{311\}$
[013]	$\frac{1}{2}\{311\}$

Figure R2. Extra reflections from FCC planar defects that can lead to diffuse signals in the SAED patterns. Figure reproduced from Ref³.

Other approaches to characterize SRO, such as the atomic-resolution scanning transmission electron microscopy energy dispersive x-ray spectroscopy (STEM-EDX)², have their own set of limitations, including issues with sample thickness and the potential mischaracterization of local element variations. For instance, STEM-EDX images provide a projected view of the actual sample, obscuring details along the thickness direction (or projection direction). Given that the thickness of a TEM sample (50 - 100 nm) significantly exceeds the size of SROs, the EDX results yield a volume-averaged signal. This makes pinpointing individual SROs challenging. Some scholars contend that the elemental

variations detected by previous STEM-EDX experiments reflect local clustering, rather than SRO. Currently, we believe that there are multiple origins of diffuse scattering from electron microscopy in these alloys that make it impossible to uniquely identify SRO with them, at least in CrCoNi.

More recently, *atomic resolution* electron tomography⁵ has been applied to characterize SRO in MEAs/HEAs, as shown in Fig. R3. Such an experiment involves tilting the specimen over a range of angles and acquiring images at each tilt increment. Using computational algorithms, these 2D projection images are then combined to reconstruct a 3D representation of the specimen. This new technique, although promising, is currently constrained to nanoparticles with a diameter smaller than 10 nm and involves a time-intensive and complicated data collection process (imaging at different tilt angles, each data point can take 1 day) that is not suitable for in-situ TEM nanomechanical testing that requires a fast imaging speed and a thin film sample. In addition, it has only been shown to work for elements with large atomic differences (such as the NiPdPt alloy) and is unlikely to work for CrCoNi.

Figure R3. 3D atomic structure of M/HEA nanocrystals revealed by atomic resolution electron tomography. Figures reproduced from Ref⁵.

In light of these above challenges, our study in this manuscript adopts an innovative, indirect approach to characterize SRO and its effects. We deliberately avoid methods that have stirred controversy and instead focus on understanding the dynamics of stacking faults (SFs) and the effects of SRO on SFs during deformation. Our simultaneous atomistic modeling aims to furnish more robust evidence for the presence of SRO in MEAs. We believe that this indirect methodology, in conjunction with other existing evidence, will offer a comprehensive insight into the phenomenon.

Comment 3:

Is there any experimental observation to determine SRO type and size?

Response.

Thank you for emphasizing the importance of determining SRO type and size, both of which are critical in understanding the SRO evolution.

We have tried to determine the SRO type and size in the past. The sample we used in this manuscript is exactly the same ice-water quenched sample used in a previous paper¹, in which we leveraged the diffuse signals from energy-filtered selected area electron diffraction (SAED) patterns and the corresponding energy-filtered dark field image to study the SRO size and distribution. The annealed sample shows diffuse streaks in the SAED patterns and clusters in the dark-field images, while the quenched one did not (Fig. R4). It was previously assumed that the degree of SRO in the quenched sample is substantially low, such that this energy-filtered TEM method is not able to resolve the SRO size and distribution in the sample. However recent work has demonstrated that even in fast-cooled samples some degree of diffuse scattering always exists⁶.

Figure R4. Characterization of the SRO in MEA using the energy-filtered. Figures reproduced from Ref¹.

Similarly, our work in this manuscript shows that even in the ice-water quenched sample, there is still a significant degree of SRO, leading to relatively high stacking energy before the mechanical deformation. As such, the deformation process around the crack tip involves two critical steps, including (1) the local destruction of SRO by dislocation glide; and (2) the irreversible extension of SFs / twins. If the initial degree of SRO is low, we can only observe the 2nd step. Thus, one of the important implications of our experiment is that SRO is ubiquitous in MEAs, even in the in ice-quenched samples. This agrees with a previous publication that shows that SRO can exist in a splat-quenched sample formed at an ultra-high cooling rate⁶.

It should be noted that we did try to see if we could identify regions of SRO during the in-situ TEM mechanical experiments with 4D-STEM. However, we found it significantly challenging. Most likely, the 4D-STEM mode used for our in-situ TEM experiments is not suitable for SRO imaging as the probe size is 1 nm, which is the smallest probe possible to still have parallel illumination necessary to interpret the diffraction patterns. On the other hand, the characterization of SRO in our previous paper¹ used the Zeiss LIBRA 200MC TEM, while none of its TEM holders can fit our push-to-pull (PTP) samples. Although we can lift out the sample from the PTP using focused ion beam (FIB) and transfer it to a sample grid that fits in the LIBRA 200MC TEM, this process inevitably destroys/changes the sample due to FIB damage or contamination.

Last but not least, we hope to highlight the challenges faced in this domain and acknowledge the various methodologies and their respective limitations.

- 1) **SRO size and its distribution**: Currently, the most widely used method for the determination of SRO size and its distribution is using dark field images generated from the diffuse spots in the selected area electron diffraction. However, there exists significant controversy. As highlighted in recent articles on *Nature Materials*³ and *Nature*⁴, the origins of the diffuse spots in the SAED pattern might arise from planar defects or HOLZs rather than SRO. Furthermore, even if the diffuse signals are indeed from SRO, the 2D projection nature of dark field TEM images poses challenges for analyzing the SRO size and distributions, especially when the sample thickness greatly exceeds the SRO size (see our schematic drawing in Fig. R5 and our comment about STEM-EDX in our reply to your comment 2). This discrepancy can lead to misleading interpretations of SRO size due to the overlap in the projection direction.

Figure R5. Schematic drawing showing the limitation of using a TEM image to determine the SRO size and distribution when the sample thickness is much larger than the SRO size.

- 2) **SRO type:** Regarding SRO type determination, the closeness in atomic numbers and sizes of Ni, Cr, and Co adds to the complexity of differentiating local chemical orderings, even with the most advanced electron tomography. Noteworthy efforts to characterize SRO type via electron tomography⁵ have utilized samples that contain elements with considerable z differences, such as Ni and Pt (Fig. R3). However, this approach may not be universally applicable, especially for NiCoCr MEA where the atom sizes are quite similar. Meanwhile, atomic resolution EDX has been employed to discern chemical ordering types. Yet, it brings forth its own set of challenges, as previously discussed in the reply to your comment 2, notably the issues related to sample thickness and the potential confusion with local clustering.

We completely concur with the emphasis on characterizing SRO size and type. As of now, obtaining reliable methods for this characterization has posed significant challenges. Despite our desire to provide detailed insights into the SRO size and type, current techniques fall short of offering unequivocal results. Thus, offering the SRO size and type evolution during in-situ TEM mechanical testing is beyond the scope of current work. However, we remain optimistic about the future and are keen on venturing further in this direction, hoping to develop or adopt new methodologies that can shed more light on this aspect.

It is worth noting that while the precise determination of SRO size and type would certainly augment our understanding, we believe the absence of this specific information does not undermine the core findings of our study. Our research primarily underscores the pivotal role SRO plays in the mechanical deformation process. Specifically, we emphasize that

SRO functions as a switch for various deformation mechanisms, laying the foundation for the exceptional mechanical performance observed in MEA/HEAs. We appreciate your insights, and they will be instrumental in guiding our future endeavors in this domain.

Comment 4:

Even though SRO can be destructed by dislocations, small-sized SRO after cutting by dislocations can remain in the system. Hence, the evolution in the density and size of SRO with and without deformation should be clarified.

Response.

SRO in MEAs/HEAs refers to the preferential bonding of elements in the nearest neighbor shells, so the SRO map should be diffuse contours, such as what we have shown in Fig. 5 in our manuscript main text. We think the SRO size is referring to the size of the cluster of regions with similar degree of SRO. In this vein, we agree with the points raised by the reviewer that “Even though SRO can be destructed by dislocations, small-sized SRO after cutting by dislocations can remain in the system.”

During cyclic loading, dislocations predominantly operate on specific glide planes. As these dislocations glide on their activated glide planes, they invariably disrupt the SROs located on those planes. This action can fragment a larger SRO cluster, leading to a decrease in the average SRO size but simultaneously increasing the local density of these smaller clusters. Moreover, certain SROs, primarily the smaller ones, which do not intersect with an activated glide plane, remain undisturbed post-loading. The above processes are schematically illustrated in Fig. R6 below.

Figure R6. Schematic drawing showing the influence of dislocation glide on SRO cluster/region density and size.

To improve our manuscript, we have incorporated this schematic representation in the supplementary materials, with an additional paragraph in the main text to provide a thorough explanation (See Fig. R7).

It should be noted that during cyclic loading, dislocations predominantly operate on specific glide planes. As these dislocations glide on their activated glide planes, they invariably disrupt the SRO regions located on those planes. This action can fragment a larger SRO region, leading to a decrease in the average SRO region size but simultaneously increasing the local density of these smaller regions. Moreover, certain regions of SRO, primarily the smaller ones, which do not intersect with an activated glide plane, remain undisturbed post-loading (Supplementary Fig. S6).

Figure 7. Update the manuscript on page 8 to address comment 4 of reviewer 1.

We believe these additions will provide clarity on the behavior and transformation of SRO regions during mechanical deformation and appreciate the insightful comment that led us to refine our manuscript in this aspect. Regarding the measurement of the evolution in the density and size of SRO with and without deformation, as mentioned in our response to comments #2-3, there currently does not exist a reliable method for direct measurement of SRO. Nevertheless, we remain hopeful that advancements in imaging techniques will soon allow us to empirically validate these phenomena.

Comment 5:

The MD simulation strain rate (typically, $10^7 \sim 10^9 \text{ s}^{-1}$) is several order of magnitude higher than the experimental one. In that case, how to prove that the simulation predictions are able to support the experimental observations?

Response.

Your concerns regarding the discrepancy between the strain rates in MD simulations and experimental conditions are well-founded and commonly raised in integrated experimental-modeling studies. Indeed, MD simulations, due to their inherent computational constraints, operate at significantly elevated strain rates compared to experimental setups. This discrepancy can lead to reservations about the direct translatability of simulation results to experimental observations.

However, it is crucial to highlight that the primary value of simulations, in this context, is not to replicate the exact experimental conditions but to capture the underlying physics and mechanisms of the processes under investigation. Our study primarily focuses on the evolution and interactions between stacking faults (SF) and SRO during cyclic deformation. Our model captures the intricate dynamics between leading and trailing

partial dislocations and the subsequent SF formation. While the strain rate in the simulations is significantly elevated, the essential physical processes and mechanisms remain consistent. Furthermore, our prior studies on MEAs deformed at a high strain rate ($\sim 10^7$ to $10^8/s$, similar to that of MD yielded) in laser-driven shock experiments⁷ showed deformation microstructures similar to our experiments at lower strain rates (i.e., profuse planar defects including stacking faults and nanotwins, see Fig. R8), reinforcing the relevance of our current simulations.

Fig. R8. Deformation microstructure of a CrCoNi MEA subjected to extreme shock loading. Figure reproduced from Ref. ⁷.

Comment 6:

Why did the authors select the pure Ni single crystal for comparison, but not select polycrystal Ni?

Response.

Thanks for pointing this out. The bulk MEA sample is polycrystalline with a large grain size, while the bulk Ni sample is a single crystal. At the size scale of our TEM samples, both MEA and Ni samples are single crystals with the same orientation. We prefer single crystals for in-situ TEM experiments because we want to exclude the effects of orientation and grain boundaries in our current study to generate more reliable results. We have added words in our manuscript to clarify this confusion (see Fig. R9).

we have developed a new characterization technique, by combining nanomechanical testing, *in situ* energy-filtered 4D-STEM, and advanced defect classification algorithms, as illustrated in Fig. 1. This approach enables us to simultaneously tackle the limitations of previous methods and provide a comprehensive understanding of the evolution of SFs and TBs in MEA near a crack tip during deformation. A water-quenched CrCoNi MEA processing SRO and a pure Ni single crystal sample used for comparison, were selected for the study. The samples were firstly twin-jet electrochemically polished to be electron transparent with [110] as the surface normal, then they were lifted out by focused ion beam (FIB) and transferred to push-to-pull (PTP) microelectromechanical system (MEMS) chips, as illustrated in the scanning electron microscopy (SEM) images in Fig. 1a-b. Both the MEA and the pure Ni sample transferred to the MEMS chip are single crystals with the same orientation. The thin samples were further patterned by FIB to have smooth fillets at the sides and a sharp crack at the center, as shown in Fig. 1c. This setup enables uniaxial tensile testing of the thin specimen in the TEM. During the FIB transfer and patterning process, we used a special method that prevents ion beam damage and Ga contamination in the region of interest in the sample (see Method for more details). In a 4D-STEM experiment,

Figure R9. Update of the manuscript on page 4 to address comment 6 of reviewer 1.

Comment 7:

When the temperature is high enough, diffusion to defect formation energy can be reduced. Hence, on page 2 or page 3, this reviewer cannot agree that the defect formation energies are constant.

Response.

Thank you for pointing out this confusion. Indeed, the defect formation energy is temperature-dependent. For example, the temperature dependence of point defect formation energy is shown in Fig. R10.

Figure R10. Effect of temperature on point defect formation energy. Figure reproduced from Ref. ⁸.

We have adjusted the phrasing in our manuscript for added precision, as reflected in Fig. R11. To clarify, our primary intent was to emphasize that SRO has the ability to modulate the SF energy across a broad range, not to suggest that temperature doesn't influence defect formation energy.

Plastic deformation in crystalline materials occurs via structural changes known as defects such as dislocations, stacking faults (SFs), and twins. The reversibility¹⁷⁻¹⁹ of these defects typically depends on the defect formation energy^{20,21}. When the defect formation energy is high and positive, the system prefers to remove the defects when the external driving force is released, thus minimizing its total free energy. When the defect formation energy is relatively small positive or even negative, defects tend to maintain their original shape after the release of the external driving force, leading to irreversible changes in materials. The defect formation energy has significant implications for the deformation mechanisms and mechanical performance in materials^{20,22}. For traditional alloys where there are only one or two principal elements with several trace elements, the defect formation energies are usually fixed **when the temperature is fixed**. Recent studies^{6,11} of HEA/MEAs have expanded this view, as theoretical studies showed that SRO can tune the stacking fault energy²³ in a wide range from negative to positive in the same material, stabilizing heterogeneous structures at the nanoscale with the co-existence of multiple dominating

Figure R11. Update of the manuscript on page 3 to address comment 7 of reviewer 1.

Comment 8:

The microstructural discrepancies between the experimental sample and the simulation model. The experimental CrCoNi sample is polycrystalline, while the simulated sample is single crystalline. A bi-/poly-crystal model with grain boundaries is highly recommended to add for comparison.

Response.

Thank you for the insightful suggestion. As addressed in our response to your Comment 6, both the CrCoNi and MEA samples used in our experiments are single crystals at the size scale of our TEM experiments, aligning with our simulations which predominantly study single crystals.

Our decision to concentrate on single crystals stems from a specific focus on the effects of deformation on the SRO and the reversibility of SF. Starting with a model system of a singular grain aids in isolating variables. We recognize the significance of grain boundaries and intend to explore their effects in forthcoming research.

Comment 9:

On page 6, what is the meaning of degradation of ordering? Is it the destruction of SRO? Or, reduced SRO degree? It is hard for general readers to understand.

Response.

We have changed “degradation of SRO” to “reduction of the degree of SRO” (Fig. R12). Thank you for your great suggestion.

to explore the interactions between dislocations and SRO. The snapshots in Fig. 4a-c show the evolution of deformation in front of the crack tip, including dislocation nucleation, SFs formation, and twinning. The FCC and HCP phases are colored green and red, respectively. The formation of SFs is highly reversible in the first 5-7 cycles, indicating a positive SF energy, as shown in Movie S3 and Fig. 4. During the cyclic deformation, dislocation activity and SFs formation occur close to the crack tip region, leading to a decrease in SRO as the cycles number increases, as shown in Fig. 4d. The reduction of the degree of SRO due to deformation leads to the continuous reduction of general SF energy near the crack tip. This continuous reduction of SRO during cyclic loading also suggests that the reversible SF extension and retraction are pseudoelastic by nature, as the glide of trailing partials and leading partials will inevitably distort the SRO arrangement locally. After around 20 cycles in the simulations, the SF length started to increase dramatically, indicating that the SF energy has dropped to a threshold value, below which SFs become a dominating

Figure R12. Update of the manuscript on page 7 to address comment 9 of reviewer 1.

Comment 10:

There are many typos, such as full name of MD in the main text on page 3 and evolution of SFs TBs on page 3. These errors are careless.

Response.

We would like to thank Review #1 for providing helpful comments. We have corrected the typos you mentioned. Please find the corrections in Fig. R13 and R14 below. Some other typos are also corrected.

To address this crucial knowledge gap, it is imperative to in-situ probe the evolution of SFs and TBs in HEA/MEAs under controlled straining conditions, with a combination of high spatial resolution to resolve individual defects and a wide field of view to enable accurate statistical analysis. Nevertheless, previous techniques^{17,19,29} such as high-resolution transmission electron microscopy (TEM), TEM dark-field imaging, or electron channeling contrast imaging (ECCI) encounter various challenges, such as a restricted field of view, a high rate of radiation damage, a

Page 3 of 25

Figure R13. Update of the manuscript to address comment 10 of reviewer 1.

Molecular dynamics simulation

MD simulations were performed using the software package LAMMPS³⁶ and the atomic configurations were displayed by OVITO³⁷. The Embedded Atom Model (EAM) potential for CrCoNi was used to describe the interatomic interactions¹¹. The dimension of simulation cells illustrated in Fig. 4 is around 800 Å in x , 600 Å in y and 50 Å in z . The Monte Carlo MD (MC/MD) simulations were first conducted to equilibrate the SRO in the sample before the deformation was applied. During the MC/MD simulation, the periodic boundary conditions were set for all three

Figure R14. Update of the manuscript to address comment 10 of reviewer 1.

Reviewer: 2

Comment 1:

This manuscript addresses an important role of short-range order (SRO) in Medium Entropy Alloys (MEA) during the deformation process by investigating the significant increase of stacking fault (SF) length after thousands of loading cycles via four dimensional scanning transmission electron microscopy (4D-STEM). In reviewer opinion, the manuscript needs a major revision before it could be considered again for Nature Communications. The authors may wish to consider the following points.

Response.

We appreciate the suggestion for revision of the manuscript and the helpful and encouraging comments by reviewer #2.

Comment 2:

The connection between experimental observation with molecular dynamic modelling support needs a clarification. for example, can the EAM potential provides any data for the SF energy of 13.3 mJ/m² for CrCoNi MEAs?

Response.

Thank you for emphasizing the need for clarity regarding the bridge between our experimental findings and MD modeling. The EAM potential⁹ that we employed originates from the work of Evan Ma and Qing-jie Li et al. Their publication, which outlines the methodology behind this potential, also presents various scientific outcomes derived from simulations using this very potential. Notably, they probed the influence of annealing on SF energy, considering that the extent of SRO can be modulated by annealing, and how the SRO can regulate SF energy over a vast spectrum. A pertinent result is illustrated in Fig. R14, wherein a sample annealed approximately at 1150 K exhibited an SF energy that aligns closely with our experimental observation of 13.3 mJ/m².

It is crucial to mention that this potential has its limitations in capturing all physical processes correctly, just like many other MD potentials. Nonetheless, this MD potential adeptly captures the pivotal physics process of MEA in our experiment, emphasizing that a higher degree of SRO corresponds to a higher SF energy. Therefore, in our assessment, this potential proves apt for our study, offering valuable insights.

Figure R15. Effect of annealing temperature on the complex SF energy (CSFE). Figure reproduced from Ref. ⁹.

Comment 3:

It is also not clear for general readers at which temperature and for which composition of MEAs the authors performed in their experimental conditions.

Response.

Thank you for highlighting the need for better clarity regarding the experimental conditions of our study. We have conducted all in-situ deformation experiments at **room temperature**. The material under investigation is an **equiatomic** CrCoNi medium entropy alloy (MEA), and it is exactly the same quenched sample utilized in prior research¹.

To ensure this information is explicitly clear to all readers, we have amended the manuscript, accordingly, as shown in Fig. R16 and R17 below.

High or medium- entropy alloys (HEAs/MEAs) are multi-principal element alloys with equal atomic elemental composition, some of which have shown record-breaking mechanical performance¹⁻⁵, including the highest fracture ~~toughnesses~~ ever recorded⁴. Early studies assumed HEAs/MEAs such as CrCoNi were random solid solutions (RSS), but more recent work⁶ has centered on the role of short range ordering (SRO)⁷⁻⁹ in stabilizing the face-centered cubic (FCC) phase. However, the link between SRO and the exceptional mechanical properties of these alloys has remained elusive. The local destruction of SRO¹⁰ by dislocation glide has been predicted to lead to a rejuvenated state with increased entropy and free energy, creating softer zones within the matrix and planar fault boundaries that enhance the ductility¹¹, but has not been verified. Here, we combine *in situ* nanomechanical testing¹²⁻¹⁴ with energy-filtered four-dimensional scanning transmission electron microscopy (4D-STEM)^{15,16} and directly observe the rejuvenation during cyclic mechanical loading in **single crystal equiatomic CrCoNi at room temperature**. Surprisingly, we found that the formation of stacking faults (SFs) and twin boundaries (TBs) are completely reversible during the first loading cycle. However, after a thousand cycles, this process becomes irreversible, indicating a significant reduction of the SF energy and, thus, a rejuvenated state. This is distinctly different from an identical experiment performed on pure nickel, where the intrinsic SF energy remains unchanged throughout the deformation process. Molecular dynamics (MD) simulation further revealed that the local breakdown of SRO in the MEA is the cause of these dynamic changes in the SF reversibility. As a result, the deformation features in HEA/MEA remain planar and highly localized to the rejuvenated planes, leading to the superior damage tolerance characteristic in this class of alloys.

Figure R16. Update of the abstract to address comment 3 of reviewer 2.

low spatial resolution, or the inability to distinguish between different types of planar defects. Here, we have developed a new characterization technique, by combining nanomechanical testing, *in situ* energy-filtered 4D-STEM, and advanced defect classification algorithms, as illustrated in Fig. 1. This approach enables us to simultaneously tackle the limitations of previous methods and provide a comprehensive understanding of the evolution of SFs and TBs in MEA near a crack tip during deformation **at room temperature**. A water-quenched **equiatomic** CrCoNi MEA processing SRO and a pure Ni single crystal sample used for comparison, were selected for the study. The samples were firstly twin-jet electrochemically polished to be electron transparent with [110] as the surface normal, then they were lifted out by focused ion beam (FIB) and transferred to push-

Figure R17. Update of the manuscript on page 4 to address comment 3 of reviewer 2.

Comment 4:

The SRO parameters are very sensitive to both alloy composition as well as function of temperature and I am wondering whether the main conclusion of this work could be applied for the case where the MEAs composition is not equiatomic (although the authors didn't mention, I assume that they were looking at equal atomic composition).

Response.

Thank you for your inquiry regarding the sensitivity of short-range order (SRO) parameters to alloy composition and temperature. In this study, we indeed focused on an equiatomic medium entropy alloy (MEA). As shown in our reply to your comment 3, we have added clarification of the composition of the MEA in our manuscript.

We acknowledge that SRO parameters can be influenced by alloy composition and the specifics of thermal processing (such as annealing temperature and duration). Despite our study's focus on equiatomic MEAs, we propose that the fundamental mechanisms we have identified—namely, the dislocation-mediated disruption of SRO and its consequent effects on stacking fault energy—would exhibit similar trends in non-equiatomic MEAs/HEAs. To reflect the potential broader implications of our findings, we have expanded our discussion in the manuscript to include this perspective (Fig. R18).

While this study focuses on an equiatomic CrCoNi MEA, the fundamental mechanisms we have identified—namely, the dislocation-mediated disruption of SRO and its consequent effects on stacking fault energy—would exhibit similar trends in non-equiatomic MEAs/HEAs.

The observation of rejuvenation in a quenched MEA after cyclic destruction of SRO offers critical insights into the impact of SRO on the mechanical performance of HEA/MEAs, giving us a greater understanding of why certain HEA/MEAs shows a combination of high strength and ductility. Previous MD simulations have shown that SRO in the CrCoNi MEA can lead to a phenomenal increase in yield strength^{11,35}. Furthermore, it is revealed by recent studies that SRO can promote work hardening and thus enhance ductility⁸. Our observations here suggest that the spontaneous local rejuvenation and softening due to the local breakdown of SRO has a dramatic effect on the local structure that is beneficial for fracture toughness. The local rejuvenation in

Figure R18. Update of the manuscript on page 9 to address comment 4 of reviewer 2.

The investigation into how SRO parameters fluctuate with alloy composition is a dynamic and active area of research. While this particular aspect is not within the scope of our current study, we are cognizant of its significance in the field. For those interested in the ongoing developments, we recommend following the work of Professor Bi-Cheng Zhou from the University of Virginia. At the 2023 Gordon Research Conference on Physical Metallurgy, Professor Zhou presented a novel computational thermodynamics model that incorporates SRO into CALPHAD calculations. This model was demonstrated with a

ternary phase diagram that predicts SRO variations across different compositions. While their findings are yet to be formally published, a preprint detailing their methodology is available on arXiv at <https://arxiv.org/pdf/2306.15384.pdf>.

We believe that both our research and these emerging studies contribute to a comprehensive understanding of SRO behaviors across a spectrum of MEA compositions.

Comment 5:

Note also that the MD simulations were performed at 800K where the random solid solution for this alloys may be dominant rather than SRO configurations as shown in Fig. b from Ref.

Response.

Thank you for your comment. We would like to clarify the methodology and rationale behind our simulations.

Our simulation procedure consists of two principal steps:

1. Hybrid molecular dynamics (MD) and Monte Carlo (MC) simulations to establish the structure of the CrCoNi sample with SRO at **800 K**. We commence with an MD simulation in the NPT ensemble to equilibrate the system at 800 K and zero pressure. This is followed by MC steps consisting of attempted atom swaps at same temperature, hybrid with the MD.
2. Mechanical deformation simulations at **300 K**.

It is important to note that only the first step is performed at 800 K. This step uses a methodology similar to the hybrid MC/MD simulations in the reference by Li et al⁹. Their findings (See Fig. R19), which we assume include Figure b referenced in your comment, demonstrate that the SRO is stronger if annealed at lower temperatures. At 800 K, the SRO is not at its peak, yet it is still notably significant. Moreover, SRO remains present even near the melting point, as indicated by non-zero SRO parameters (Fig. R19a). This observation is corroborated by recent experiments showing the presence of SRO in splat-quenched samples formed under ultra-high cooling rates⁶.

The choice of 800 K for the annealing step is deliberate, mirroring one of the common experimental annealing temperatures for MEAs/HEAs. Lower annealing temperatures would indeed result in a greater degree of SRO; however, the kinetics at such temperatures are substantially slower, and the time required to reach the converged SRO parameters would be impractically lengthy for real-world conditions.

Figure R19. Effect of annealing temperature on the degree of SRO. Figure reproduced and modified from Ref. 9.

Comment 6:

The authors showed in Figure 4d the destruction of SRO after cycle loading in term of so-called overall SRO dependence. However, by looking at the different SRO pair dependence in Fig. 4e, it is surprise for me to see the presence of negative SRO trend for Ni-Ni pairs that is contradicting to the positive SRO value of Ni-Ni from Fig. 1b in Ref. 11. Note from the same reference the SRO of Cr-Cr are slightly negative whereas in the present study the authors showed that the are strongly positive.

Response.

We appreciate your observation regarding the discrepancy in the short-range order (SRO) trends reported in our study compared to those in Figure 1b of Ref. 11. In our study, we utilized the molecular dynamics (MD) simulation potential and the Monte Carlo (MC)/MD annealing methodology from Ref. 11, but we opted for a different approach in defining the SRO parameters.

The SRO parameters in our study are based on the traditional Warren-Cowley parameters¹⁰, which are more widely recognized within the field. In contrast, Ref. 11 employs a modified version of these parameters. We believe that adhering to the conventional definition will minimize confusion and align more closely with the majority of existing literature on the topic.

To elucidate this distinction, we have provided a comparison between the traditional Warren-Cowley SRO parameters and the modified version used in Ref. 11:

1. Warren-Cowley parameters (used in our study):

$$\alpha_{ij} = 1 - \frac{P_{ij}}{c_j}$$

where P_{ij} is the fraction of species j in the nearest-neighboring shell around i , and c_j is the concentration of j . Positive values of the Warren-Cowley parameter indicate an unfavored tendency of atomic pairs, whereas negative values suggest a favored tendency.

2. Modified Warren-Cowley parameters⁹ (used in Ref. 11):

$$\alpha_{ij} = \frac{P_{ij} - c_j}{\delta_{ij} - c_j}$$

where P_{ij} is the fraction of species j in the nearest-neighboring shell around i , c_j is the concentration of j , and δ_{ij} is Kronecker delta function:

$$\delta_{ij} = \begin{cases} 0 & \text{if } i \neq j \\ 1 & \text{if } i = j \end{cases}$$

For pairs with the same species (i.e., $i = j$), a positive α_{ij} suggests preferred segregation. However, when $i \neq j$, a negative α_{ij} suggests j -type clustering the specific shell of an i -type atom.

Upon comparing the two methodologies, we have observed that for cases where $i \neq j$, both approaches yield equivalent results. However, when $i = j$, these two methods will lead to not only opposite signs but also different absolute values. Thus, a direct comparison of our results with those from Ref. 11 is most effectively done by examining the cases where $i \neq j$. In our paper, $\alpha_{Co-Cr} \approx -0.29$, while $\alpha_{Co-Cr} \approx -0.35$ in Ref. 11. These values are in close agreement. The absolute value of the SRO is contingent on the selection of MC/MD simulation parameters and is not a critical factor for our study since our primary objective is to observe the changes in SRO due to mechanical deformation.

Reviewer: 3**Comment 1:**

The paper illustrates the mechanical rejuvenation of CrCoNi MEA induced by cyclic loading through an in-situ nanomechanical testing method, along with observations using the newly developed energy-filtered 4D STEM technique. The authors directly observe the evolution of defects, such as stacking faults and twins, during cyclic loading. This observation suggests a reduction in stacking fault energy due to the rejuvenation process, which could stem from the destruction of short-range order (SRO) caused by dislocation activities in the crack tip region. The innovative experimental method carries significant potential and could broadly impact nanomechanics studies, contributing to a more profound understanding of materials' fracture and deformation phenomena. However before considering publication, the following points can be addressed.

Response.

We are grateful for your positive and insightful feedback regarding our manuscript. Your recognition of the innovative nature of our experimental method and its potential broad impact on the field of nanomechanics is highly encouraging.

We are eager to address the points you have raised and believe that through addressing these, our paper will be significantly improved and ready for publication. Your constructive feedback is invaluable in assisting us to enhance the clarity and depth of our study. Thank you for contributing to the refinement of our work.

Comment 2:

The mechanism of the transition from reversible to irreversible behavior at certain cyclic loadings should be presented and discussed more clearly. This mechanism is the foundation for the destruction of SRO and the formation of fine, soft (rejuvenated) and hard (chemically ordered) zones at the crack tip.

Response.

We concur that the transition from reversible to irreversible stacking fault (SF) behavior during cyclic loading is pivotal, and its mechanism is essential for understanding the SRO-dislocation interactions that govern this process. We appreciate the need for greater clarity on this topic and have revised our discussion to elucidate these mechanisms.

The transition can be interpreted from two perspectives:

Firstly, from a free energy standpoint, the SF energy is a critical determinant of SF stability—the lower the SF energy, the more stable the SF. The reversible to irreversible transitions suggest a corresponding shift from higher to lower stacking fault energy. We postulate that this shift results primarily from the dislocation glide disrupting the SRO, thereby progressively reducing the SF energy and softening the glide planes. This has been verified by our simulations.

Secondly, we may benefit from the view of the SF length. A SF in FCC materials is the area between two Shockley partial dislocations. The length of the SF depends on how far the leading partial can travel. When the degree of SRO is high, the SRO can retard the glide of the leading partial dislocations, and because SF energy is high, the partial would like to retract after the release of the external load. Thus, the SF is reversible. Conversely, when SRO is diminished, dislocations encounter less resistance and can traverse further, resulting in longer stacking faults that persist even after the external load removal due to the reduced stacking fault energy.

It should be noted that during cyclic loading, dislocations predominantly operate on specific glide planes. As these dislocations glide on their activated glide planes, they invariably disrupt the SRO regions located on those planes. This action can fragment a larger SRO region, leading to a decrease in the average SRO size but simultaneously increasing the local density of these smaller regions. Moreover, certain SROs, primarily the smaller ones, which do not intersect with an activated glide plane, remain undisturbed post-loading. The above processes are schematically illustrated in Fig. R6 below.

We have added the above discussions in our manuscript, as highlighted in Fig. R20 below.

Figure R6. Schematic drawing showing the influence of dislocation glide on SRO region density and size.

Discussion and conclusion

The transition from reversible to irreversible stacking fault (SF) behavior during cyclic loading is pivotal, and its mechanism is essential for understanding the SRO-dislocation interactions that govern the rejuvenation process. The transition can be interpreted from two perspectives.

Firstly, from a free energy standpoint, the SF energy is a critical determinant of SF stability—the lower the SF energy, the more stable the SF. The reversible to irreversible transitions suggest a corresponding shift from higher to lower stacking fault energy. In this work, we first postulate that this shift results primarily from the dislocation glide disrupting the SRO, thereby progressively reducing the SF energy and softening the glide planes. This has been then verified by our MD simulations.

Secondly, we may benefit from the view of the SF length. A SF in FCC materials is the area between two Shockley partial dislocations. The length of the SF depends on how far the leading partial can travel. When the degree of SRO is high, the SRO can retard the glide of the leading partial dislocations, and because SF energy is high, the partial would like to retract after the release of the external load. Thus, the SF is reversible. Conversely, when SRO is diminished, dislocations encounter less resistance and can traverse further, resulting in longer stacking faults that persist even after the external load removal due to the reduced stacking fault energy.

It should be noted that during cyclic loading, dislocations predominantly operate on specific glide planes. As these dislocations glide on their activated glide planes, they invariably disrupt the SRO regions located on those planes. This action can fragment a larger SRO region, leading to a decrease in the average SRO region size but simultaneously increasing the local density of these smaller regions. Moreover, certain regions of SRO, primarily the smaller ones, which do not intersect with an activated glide plane, remain undisturbed post-loading (Supplementary Fig. S6).

Figure R20. Update of the manuscript on page 8 to address comment 2 of reviewer 3.

Comment 3:

Exploring the transition in relation and the evolution of stacking fault length to cyclic loading frequency and strain amplitude could be intriguing.

Response.

We appreciate your suggestion to investigate the effects of loading frequency and strain amplitude on the interaction between dislocations, stacking faults (SFs), and short-range order (SRO). While the idea is indeed intriguing, it falls outside the scope of the current paper, which focuses on the interaction dynamics under experimentally practical and convenient loading conditions.

We would also like to draw attention to the practical experimental challenges associated with studying these effects at the nanoscale. The sample preparation process—twin-jet-polishing followed by focused ion beam (FIB) patterning—inevitably results in variability between samples, such as differences in thickness and crack geometry. These variances can significantly affect the local strain rate and stress intensity factors at the crack tip, even when the external load and frequency remain constant. Given the current precision limitations of FIB technology, it is challenging to conduct a quantitatively precise study on

the impact of loading frequency and strain amplitude on samples such as this. Despite these challenges, we recognize the value of such a study and hope that future advancements in nanomechanical testing will enable us to explore these effects more thoroughly.

Comment 4:

Also, addressing the possibility of vacancy and interstitial generation during cyclic loading and its impact on defect evolution and stacking fault length would be beneficial.

Response.

We acknowledge your suggestion regarding the role of vacancies and interstitials generated during cyclic loading. Indeed, this is a complex but crucial subject, and thus it represents an actively evolving research area. While a comprehensive study of them is beyond the scope of this work, we hope to briefly touch upon the mechanisms at play that are relevant to this study, *i.e.*, how vacancies generated during cyclic loading can influence short-range order (SRO) evolution even at room temperature.

1) Vacancy facilitates the formation of SRO.

During cyclic loading, vacancies will be emitted from the crack tip¹¹ directly, or generated by the back-and-forth dislocation movement even at room temperature¹². These excess vacancies can facilitate diffusion. Based on the modeling⁹ by Li et al (Fig. R19, see below), the degree of SRO is higher when the annealing temperature is lower. Thus, at room temperature, the thermal dynamic driving force for SRO formation is very high. However, SRO cannot reach this high SRO value predicted at its equilibrium at room temperature because the kinetics is typically too slow. The excess vacancy led by cyclic loading may facilitate SRO formation due to more rapid kinetics.

Figure R19. Effect of annealing temperature on the degree of SRO. Figure reproduced and modified from Ref. ⁹.

Previously, people have studied how precipitate or nanocluster interact with vacancies and dislocations during cyclic loading, such in aluminum alloys¹².

For example, based on a classical (Zener) model for diffusion-controlled growth of spherical particles, the growth rate of particle has an inverse relationship with the strain rate. As shown in previous studies¹², the growth rate of local clusters during cyclic deformation comprises a destructive term tied to dislocations and a constructive term associated with vacancy-mediated mechanisms, the latter being highly sensitive to the strain rate, as shown by the equation below¹²:

$$\frac{dR}{d\epsilon} = \frac{dR^+}{d\epsilon} + \frac{dR^-}{d\epsilon}$$

where R and ϵ are the size of the spherical particle and the strain, respectively. For the Zener model, the gain term can be:

$$\frac{dR^+}{d\epsilon} = \frac{1}{\dot{\epsilon}} \frac{dR}{dt} = \frac{1}{\dot{\epsilon}} \left[\frac{D \cdot (C_b - C_{eq})}{R \cdot (C_p - C_{eq})} \right] \frac{C_{exv}}{C_{eqv}}$$

where C_b , C_p , C_{eq} , C_{eqv} , C_{exv} , t , $\dot{\epsilon}$, and D are bulk solute content, solution concentration in the precipitate, equilibrium solute concentration, equilibrium vacancy concentration, excess concentration of vacancy, time, strain rate and diffusivity of the solute in the presence of an equilibrium concentration of vacancies, respectively.

While the model for spherical precipitate or local cluster may not be directly applied for the SRO, it was hypothesized that the mechanisms might be similar. The vacancy-mediated SRO formation, though crucial in systems at lower strain rates, gets significantly diminished in high-strain-rate scenarios.

At lower strain rates and over extended cycling periods, the influence of vacancies becomes more pronounced. These excess vacancies may decelerate the dislocation-mediated destruction of SRO or potentially enhance the degree of SRO. Consequently, dislocation motion is expected to become more sluggish due to more SROs acting as an impediment. Furthermore, with a higher degree of SRO, the SF energy increases, resulting in fewer long and irreversible SFs. This leads to a tendency for SFs to be more transient and reversible in nature.

For our system, given the high melting temperature (1690 K) of CrCoNi MEA¹³, the homologous temperature is low ($T/T_m = 0.178$), and the diffusion processes are inherently sluggish. Thus, the vacancy-mediated SRO formation is considered minimal and possibly negligible. Indeed, our observations indicate a continuous

reduction in SF energy throughout our experiments, which suggests a diminishing degree of SRO. This trend points to the predominance of dislocation-induced SRO destruction in the deformation process. Additional evidence is the electric resistivity¹⁴ studies on CrCoNi MEA under single-cycle tensile loading at a similar strain rate, which also shows the dominance of SRO reduction rather than SRO formation (Fig. R21).

Figure R21. Variation of the electrical resistivity for CrCoNi as a function of engineering strain during tensile testing. Figure modified from Ref.¹⁴.

- 2) Other effects of vacancies and vacancy clusters on dislocation include altering dislocation mobility, facilitating dislocation climb, and lock or pinning of dislocation, etc. At room temperature, these interactions are generally less dynamic than at higher temperatures, owing to reduced atomic mobility.

We have included the above discussions in our manuscript and supplementary materials, as shown in Fig. R22 – R23 below.

In addition, it is worthwhile to comment on the role of excess vacancies in defect evolution during cyclic loading. Vacancies emitted during cyclic loading can facilitate diffusion and, consequently, the degree of SRO. However, as detailed in the Supplementary Text, the role of vacancy-mediated SRO formation in our experiments is considered minimal and possibly negligible. Indeed, our observations indicate a continuous reduction in SF energy, suggesting a diminishing degree of SRO. This trend points to the predominance of a dislocation-induced SRO destruction mechanism in the deformation process.

Figure R22. Update of the manuscript on pages 8-9 to address comment 4 of reviewer 3.

Figure R23. Update of the supplementary materials to address comment 4 of reviewer 3.

Comment 5:

Considering that the stacking fault energy of Ni is significantly higher compared to MEA, even with SRO, a fair comparison of planar defect evolution between Ni and MEA might be challenging. Authors could consider comparing with Cu for a more comprehensive comparison.

Response.

Thank you for your comment. We chose Ni as a reference material because it is a constituent element of the CrCoNi MEA under this study and shares the same face-centered cubic (FCC) crystal structure. Given its high and stable stacking fault energy, Ni provides a stark contrast to CrCoNi MEA, whose stacking fault energy varies widely (from -28 to 66 mJ/m² depending on the degree of SRO¹⁵).

As the reviewer points out, Cu would serve as an interesting comparison due to its typically lower stacking fault energy of around 55 mJ/m² [Ref. ¹⁶] compared to Ni's 120 mJ/m² [Ref.¹⁷]. Both Ni and Cu have fixed stacking fault energies at room temperature, which are not influenced by deformation, precluding the observation of rejuvenation and transition in deformation mechanisms (from reversible to irreversible stacking faults). It would also have been interesting to look at the same MEA processed with different amounts of SRO (through annealing and/or slow cooling). However, due to the practical constraints of instrument time and the difficulty of the experiments, we were not able to look at all of the

different combinations of samples that we would have liked to. In the future it would be interesting to study variations of the MEA and also Cu.

Previous simulations of cyclic deformation around a crack in pure Cu¹⁸ have shown significant differences from our CrCoNi MEA results. For instance, Fig. R24 presents MD simulation outcomes for a pure Cu sample with a crystal orientation similar to our CrCoNi simulations. We have analyzed Fig. R24 and plotted the variation of the estimated total SF length as a function of loading cycles. The results have been included in Fig. R25 below, which compares pure Cu with CrCoNi MEA.

In pure Cu, the total stacking fault length consistently increases rapidly with each loading cycle, except for the last data point where crack propagation and void formation become dominant (Fig. R25a). In contrast, the MEA exhibits a two-step process (Fig. R25b): the total stacking fault length increases slowly during the first 20 cycles due to the high degree of SRO and thus high stacking fault energy. When the degree of SRO significantly drops, the stacking fault length begins to increase at a much faster rate.

Figure R24. Snapshots showing the plastic deformation and crack growth during fatigue loading for a single crystal pure Cu. (a) 2nd cycle; (b) 3rd cycle; (c) 4th cycle; (d) 8th cycle; (e) 9th cycle; (f) 11th cycle. Figure reproduced from Ref. ¹⁸.

Figure R25. Comparison of the evolution of SFs during cyclic loading for different metals. (a) Pure Cu. The curve was plotted by us based on the MD snapshots in Ref. ¹⁸. (b) Equiatomic CrCoNi MEA studied in this paper.

We have added the above discussion in the manuscript (See Fig. R26-R27).

analysis. Nevertheless, previous techniques^{17,19,29} such as high-resolution transmission electron microscopy (TEM), TEM dark-field imaging, or electron channeling contrast imaging (ECCI) encounter various challenges, such as a restricted field of view, a high rate of radiation damage, a low spatial resolution, or the inability to distinguish between different types of planar defects. Here, we have developed a new characterization technique, by combining nanomechanical testing, *in situ* energy-filtered 4D-STEM, and advanced defect classification algorithms, as illustrated in Fig. 1. This approach enables us to simultaneously tackle the limitations of previous methods and provide a comprehensive understanding of the evolution of SFs and TBs in MEA near a crack tip during deformation at room temperature. A water-quenched equiatomic CrCoNi MEA processing SRO and a pure Ni single crystal sample used for comparison, were selected for the study. We chose Ni as a reference material because it is a constituent element of the CrCoNi MEA under this study and shares the same face-centered cubic (FCC) crystal structure. Given its high and stable stacking fault energy, Ni provides a stark contrast to CrCoNi MEA, whose stacking fault energy varies widely depending on the degree of SRO. The samples were firstly twin-jet electrochemically polished to be electron transparent with [110] as the surface normal, then they were lifted out by focused ion beam (FIB) and transferred to push-to-pull (PTP) microelectromechanical system

Figure R26. Update of the manuscript on page 4 to address comment 5 of reviewer 3.

Last but not least, the SF evolution during cyclic loading in MEA revealed by our MD simulation is compared with a previous MD simulation³³ of cyclic loading in pure Cu, which has a lower stacking fault energy (around 55 mJ/m²)³⁴ than pure Ni. We analyzed the snapshots of MD simulations in Ref. ³³ and qualitatively plotted the variation of total SF length as a function of loading cycles for pure Cu. The results have been included in Supplementary Fig. S5, which compares pure Cu with CrCoNi MEA. In pure Cu, the total stacking fault length consistently increases rapidly with each loading cycle, except for the last data point where crack propagation and void formation become dominant (Supplementary Fig. S5a). By contrast, MEA demonstrates a two-phase evolution characterized initially by a slow increment in SF length, which then accelerates, correlating with the progressive breakdown of SRO and rejuvenation in the material.

Figure R27. Update the manuscript on page 7 to address comment 5 of reviewer 3.

Comment 6:

The current demonstrations, both experimentally and in MD simulations, seem to be quasi 2-dimensional. Exploring a 3-dimensional scenario, such as a penny-shaped crack in a 3-dimensional sample, could lead to more complex dislocation activities. Therefore, discussing the effectiveness of single loading for rejuvenation in a 3-dimensional case could provide valuable insights.

Response.

Thank you for your insightful comment regarding the extension of our work to a 3-dimensional scenario. We acknowledge the importance and relevance of exploring 3D cases, such as a penny-shaped crack in a 3-dimensional sample, as they indeed present more complex dislocation activities compared to quasi 2-dimensional scenarios.

Our current study primarily focuses on quasi 2-dimensional models due to their typical use in in-situ TEM experiments, which only accommodate thin samples, and the more manageable computational load for modeling. This approach also aims to simplify the inherent complexity of the system, allowing us to isolate specific phenomena and understand the fundamental mechanisms at play.

However, we recognize the necessity of transitioning to a 3-dimensional model to enhance the applicability and scope of our findings. In a 3D system, more slip planes are activated, and more SRO interactions with dislocations occur, potentially increasing rejuvenation efficiency. In our quasi-2D study, rejuvenation was observed at around 20 cycles of loading, after which stacking faults (SFs) became irreversible with a high average length (Fig. R25b). In a true 3D scenario, fewer cycles may be required under similar loading conditions, and it is plausible that even a single loading could lead to the rejuvenation of MEAs/HEAs and alter deformation mechanisms. For instance, our investigation into the deformation microstructure of equiatomic CrCoNi MEA at room temperature, particularly in samples damaged by shock loading⁷, revealed profuse SFs/twins forming a 3D network (See Fig. R8 below), characteristic of rejuvenated MEA with low SF energy. This suggests significant implications for single loading in 3D, even as our study centers on cyclic loading in 2D.

Fig. R8. Deformation microstructure of a CrCoNi MEA subjected to extreme shock loading. Figure reproduced from Ref. ⁷.

It is important to note that the superior combination of ductility and strength in CrCoNi is not solely due to low SF energy but also the interplay of soft (locally rejuvenated) and hard zones (with high SF energy) within the material. In a single loading in a 3D scenario, while more glide planes may be softened (rejuvenated) than in the quasi-2D scenario, a significant volume remains unrejuvenated, particularly where dislocations are pinned or locked. Thus, the concept of composite materials comprising “soft” and “hard” zones remains valid for the 3D scenario and for single loading.

To address your suggestion, we have added several new paragraphs in our manuscript (Fig. R28) that discuss the potential implications of our quasi-2D study on single loading in 3D scenarios. We hope this addition addresses your concern and adds value to our study. We are grateful for the opportunity to enhance our manuscript with your valuable feedback.

Our current study primarily focuses on quasi 2-dimensional (2D) models due to their typical use in in-situ TEM experiments, which only accommodate thin samples, and the more manageable computational load for modeling. In a 3D system, more slip planes are activated, and more SRO interactions with dislocations occur, potentially increasing rejuvenation efficiency. In our quasi-2D study, significant rejuvenation was observed at around 20 cycles of loading, after which stacking faults (SFs) became irreversible with a high average length (Fig. 4d). In a true 3D scenario, fewer cycles may be required under similar loading conditions, and it is plausible that even a single loading could lead to the rejuvenation of MEAs/HEAs and alter deformation mechanisms. For instance, in CrCoNi samples damaged by shock loading at temperature⁷, it is found that profuse SFs/twins form a 3D network, characteristic of rejuvenated MEA with low SF energy. This suggests significant implications of rejuvenation for single loadings in 3D.

It is crucial to point out that the superior combination of ductility and strength in CrCoNi is not solely due to low SF energy but also the interplay of soft (locally rejuvenated) and hard zones (with high SF energy) within the material. In a single loading in a 3D scenario, while more glide planes may be softened (rejuvenated) than in the quasi-2D scenario, a significant volume remains unrejuvenated, particularly where dislocations are pinned or locked. Thus, the concept of composite materials comprising “soft” and “hard” zones remains valid for the 3D scenario and for single loading.

In summary, through the integration of in-situ energy-filtered 4D-STEM, nanomechanical testing, and MD modeling, our study reveals and elucidates the reversible to irreversible transitions of SF dynamics in CrCoNi MEA during cyclic deformation. This phenomenon is primarily attributed to the disruption of SRO along the glide planes, leading to localized material rejuvenation and the formation of a composite system. This system comprises 'soft' zones with diminished SRO and 'hard' zones where SRO is maintained. Intriguingly, it is the SRO that orchestrates the sequence of deformation mechanisms, fostering a synergy of multiple mechanisms and consequently enhancing the mechanical performance of CrCoNi MEAs. Our findings pave the way for future research aimed at enhancing the damage tolerance of structural alloys. Specifically, identifying strategies to optimize the degree and distribution of SRO in MEAs/HEAs could be pivotal in local rejuvenation processes, thereby enhancing material properties.

Figure R28. Update of the manuscript on pages 9-10 to address comment 6 of reviewer 3.

Reference for point-by-point reply:

1. Zhang, R. *et al.* Short-range order and its impact on the CrCoNi medium-entropy alloy. *Nature* **581**, 283–287 (2020).
2. Chen, X. *et al.* Direct observation of chemical short-range order in a medium-entropy alloy. *Nature* **592**, 712–716 (2021).
3. Walsh, F., Zhang, M., Ritchie, R. O., Minor, A. M. & Asta, M. Extra electron reflections in concentrated alloys do not necessitate short-range order. *Nat. Mater.* **22**, 926–929 (2023).
4. Coury, F. G., Miller, C. A., Field, R. & Kaufman, M. J. On the Origin of Diffuse Intensities in FCC Electron Diffraction Patterns. *Nature* **622**, (2023).
5. Moniri, S. *et al.* Three-dimensional atomic positions and local chemical order of medium- and high-entropy alloys. 1–35 (2023).
6. Zhang, M. *et al.* Determination of peak ordering in the CrCoNi medium-entropy alloy via nanoindentation. *Acta Mater.* **241**, 118380 (2022).
7. Zhao, S. *et al.* Deformation and failure of the CrCoNi medium-entropy alloy subjected to extreme shock loading. *Sci. Adv.* **9**, (2023).
8. Mendeleev, M. I. & Bokstein, B. S. Molecular dynamics study of self-diffusion in Zr. *Philos. Mag.* **90**, 637–654 (2010).
9. Li, Q. J., Sheng, H. & Ma, E. Strengthening in multi-principal element alloys with local-chemical-order roughened dislocation pathways. *Nat. Commun.* **10**, 1–11 (2019).
10. Cowley, J. M. An approximate theory of order in alloys. *Phys. Rev.* **77**, 669–675 (1950).
11. Nishimura, K. & Miyazaki, N. Molecular dynamics simulation of crack growth under cyclic loading. *Comput. Mater. Sci.* **31**, 269–278 (2004).
12. Sun, W. *et al.* Precipitation strengthening of aluminum alloys by room-temperature cyclic plasticity. *Science* **363**, 972–975 (2019).
13. Wu, Z., Bei, H., Pharr, G. M. & George, E. P. Temperature dependence of the mechanical properties of equiatomic solid solution alloys with face-centered cubic crystal structures. *Acta Mater.* **81**, 428–441 (2014).
14. Li, L. *et al.* Evolution of short-range order and its effects on the plastic deformation behavior of single crystals of the equiatomic Cr-Co-Ni medium-entropy alloy. *Acta Mater.* **243**, (2023).
15. Ding, J., Yu, Q., Asta, M. & Ritchie, R. O. Tunable stacking fault energies by tailoring local chemical order in CrCoNi medium-entropy alloys. *Proc. Natl. Acad. Sci. U. S. A.* **115**, 8919–8924 (2018).
16. Bahmanpour, H. *et al.* Effect of stacking fault energy on deformation behavior of cryo-rolled copper and copper alloys. *Mater. Sci. Eng. A* **529**, 230–236 (2011).
17. Carter, C. B. & Holmes, S. M. The stacking-fault energy of nickel. *Philos. Mag.* **35**, 1161–1171 (1977).
18. Potirniche, G. P., Horstemeyer, M. F., Gullett, P. M. & Jelinek, B. Atomistic modelling of fatigue crack growth and dislocation structuring in FCC crystals. *Proc. R. Soc. A Math. Phys. Eng. Sci.* **462**, 3707–3731 (2006).

REVIEWERS' COMMENTS

Reviewer #1 (Remarks to the Author):

All deep questions and comments are fully reflected in the revision. The response work was great. I, therefore, agree with the authors' knowledge found in this article and accept the publication in Nature Communications.

Reviewer #2 (Remarks to the Author):

The authors have responded to my comments point by point within the revised version. I am in general satisfied with their replies but would be happy to only recommend for its publication in Nature Communications in a condition that the authors could implement the two new amendments as follows.

(i) Regarding the authors reply to my comment 4 (Reviewer 2): The authors recommended the reviewer to read the unpublished paper presented by Dr. Zhou at an conference in 2023 w.r.t. to the temperature and composition dependence of SRO in MEA/HEA but the authors seem to ignore citing several important studies already published in literature on this topic. I suggest the authors to include at least more references on this topic as the following.

[1] A. Fernandez-Caballero et al., J. Phase Equilibrium and Diffusion, vol. 38, 391-403 (2017)

[2] D. Sobieraj et al., PCCP, vol. 22, 32929 (2020)

[3] M. Fedorov et al., PRB, vol. 101, 174416 (2020)

[4] O. El Atwani et al., Nature Communications, <https://doi.org/10.1038/s41467-023-38000-y>

(ii) Regarding the authors reply to my comment 4 (Reviewer 2): The authors explained that their definition of SRO is different to those of Reference 11. But they should include these explanations into the text of this manuscript as it is important for general readers of Nature Communications to understand about these differences.

When the two above amendments have been properly taken into account then the new corrected version can be accepted for its publication.

Reviewer #3 (Remarks to the Author):

The authors have diligently addressed most of my comments. Nevertheless, I cannot agree with their response to the second comment from Reviewer 3 regarding the transition mechanism from reversible to irreversible, which is crucial as it is supposed to trigger rejuvenation — a key aspect of this paper. The authors propose that this transition results from a reduction in free energy (stacking fault energy). However, if cyclic loading is precisely controlled, with uniform amplitude, the forward force could drive a leading partial to a specific position from the crack tip, thereby generating stacking faults and disrupting short-range order. Conversely, the subsequent reverse force should retract the leading partial back to its initial position, eliminating the stacking faults and restoring the short-range order. Therefore, despite achieving a state of lower free energy, the backward force can perform work to overcome the free energy difference.

An itemized list of responses to reviewers' remarks (Blue: Reviewer's comments; Black: Our response)

Reviewer: 2

Comment 1:

Regarding the authors reply to my comment 4 (Reviewer 2): The authors recommended the reviewer to read the unpublished paper presented by Dr. Zhou at an conference in 2023 w.r.t. to the temperature and composition dependence of SRO in MEA/HEA but the authors seem to ignore citing several important studies already published in literature on this topic. I suggest the authors to include at least more references on this topic as the following.

[1] A. Fernandez-Caballero et al., *J. Phase Equilibrium and Diffusion*, vol. 38, 391-403 (2017) [2] D. Sobieraj et al., *PCCP*, vol. 22, 32929 (2020)

[3] M. Fedorov et al., *PRB*, vol. 101, 174416 (2020)

[4] O. El Atwani et al., *Nature Communications*, <https://doi.org/10.1038/s41467-023-38000-y>

Response.

Thank you for suggesting these helpful references. We have added them as references 34-37 in our manuscript, as shown in the Fig. R1 below.

The SRO parameters can be influenced by alloy composition and thermal-mechanical processing³⁴⁻³⁷. While this study focuses on an equiatomic CrCoNi MEA, the fundamental mechanisms we have identified—namely, the dislocation-mediated disruption of SRO and its consequent effects on stacking fault energy—may exhibit similar trends in non-equiatomic MEAs/HEAs.

33. Bahmanpour, H. *et al.* Effect of stacking fault energy on deformation behavior of cryo-rolled copper and copper alloys. *Mater. Sci. Eng. A* **529**, 230–236 (2011).
34. Sobieraj, D. *et al.* Chemical short-range order in derivative Cr-Ta-Ti-V-W high entropy alloys from the first-principles thermodynamic study. *Phys. Chem. Chem. Phys.* **22**, 23929–23951 (2020).
35. El Atwani, O. *et al.* A quinary WTaCrVHf nanocrystalline refractory high-entropy alloy withholding extreme irradiation environments. *Nat. Commun.* **14**, 1–12 (2023).
36. Fedorov, M., Wróbel, J. S., Fernández-Caballero, A., Kurzydłowski, K. J. & Nguyen-Manh, D. Phase stability and magnetic properties in fcc Fe-Cr-Mn-Ni alloys from first-principles modeling. *Phys. Rev. B* **101**, 1–30 (2020).
37. Fernández-Caballero, A., Wróbel, J. S., Mummery, P. M. & Nguyen-Manh, D. Short-Range Order in High Entropy Alloys: Theoretical Formulation and Application to Mo-Nb-Ta-V-W System. *J. Phase Equilibria Diffus.* **38**, 391–403 (2017).

Fig. R1. Update the manuscript to address comment 1 of reviewer #2.

Comment 2:

Regarding the authors reply to my comment 4 (Reviewer 2): The authors explained that their definition of SRO is different to those of Reference 11. But they should include these explanations into the text of this manuscript as it is important for general readers of Nature Communications to understand about these differences.

Response.

Thank you for highlighting the need to add the note about the difference in the definition of the SRO parameter. We have added this note in the method section, as shown in Fig. R2.

Local chemical short-range order parameter

For the SRO, we adopted the pairwise multicomponent Warren Cowley order parameter³¹, $\alpha_{ij} = 1 - \frac{P_{ji}}{c_j}$, to quantify the SRO in each specific nearest-neighboring shell. $P_{j,i}$ is the fraction of species j in the nearest-neighboring shell around i , and c_j is the concentration of j . To indicate the overall degree of SRO, we make use of a quantity given as the sum of all the $|\alpha_{ij}|$ for all species at nearest-neighbor shell ($CSRO = \sum_{i,j} |\alpha_{ij}|$). Note that the Warren Cowley order parameter³¹ we employed differs from the Warren Cowley order parameter used by Li et al.¹³, which involves the use of the Kronecker delta function.

Fig. R2. Update the manuscript to address comment 2 of reviewer #2.

Reviewer: 3

Comment 1:

The authors have diligently addressed most of my comments. Nevertheless, I cannot agree with their response to the second comment from Reviewer 3 regarding the transition mechanism from reversible to irreversible, which is crucial as it is supposed to trigger rejuvenation — a key aspect of this paper. The authors propose that this transition results from a reduction in free energy (stacking fault energy). However, if cyclic loading is precisely controlled, with uniform amplitude, the forward force could drive a leading partial to a specific position from the crack tip, thereby generating stacking faults and disrupting short-range order. Conversely, the subsequent reverse force should retract the leading partial back to its initial position, eliminating the stacking faults and restoring the short-range order. Therefore, despite achieving a state of lower free energy, the backward force can perform work to overcome the free energy difference.

Response.

Thank you for pointing out the necessity of explaining the backward force and the possibility of restoring the short-range order. The language discussing the mechanism is further toned down in our manuscript. Also, we have discussed the “reverse force” in the main text.

We want to first clarify that in our experiment, cyclic loading uses a loading curve with forces that are always larger or equal to zero (See Fig. 2q in the manuscript). There is no negative force during the unloading stage.

Below is our understanding of the “reverse force”:

A stacking fault (SF) in FCC materials is the area between two Shockley partial dislocations. The length of the SF depends on how far the leading partial can travel. If the SF energy is positive, it induces an attractive force between the two partial dislocations. This force is typically counterbalanced by the repulsive elastic interaction between the partials, in the absence of external forces. When an external force is applied, it disrupts this balance, leading to an increase in the separation distance between the partials. Upon release of the external force, a reverse force occurs because the attractive force resulting from positive SF energy substantially outweighs the repulsive force due to the strain energy of the partials. This reverse force narrows the distance between the partials, diminishing gradually as the separation decreases (thereby reducing SF length). In a purely elastic scenario, the SF length will eventually recover to its initial state after unloading. In MPEAs, however, the scenario differs due to the presence of SRO, which may be disrupted by dislocation gliding – an inelastic process. The reduction in SRO by the leading partial alters the SF energy and therefore the reverse force upon unloading is also altered, leading to a

rejuvenated state that provides an asymmetric loading and unloading with regards to the position of the leading partial. Upon retraction, there is an energetic balance between the partial dislocation restoring the SRO at the expense of then increasing the SFE. Further studies of a pure metal with a low SFE such as Cu would be interesting for comparison in this respect.

Our changes are reflected in Fig. R3 below.

Secondly, we may benefit from the view of SF length. An SF in FCC materials is the area between two Shockley partial dislocations. The length of the SF depends on how far the leading partial can travel. If the SF energy is positive, it induces an attractive force between the two partial dislocations. This force is typically counterbalanced by the repulsive elastic interaction between the partials, in the absence of external forces. When an external force is applied, it disrupts this balance, leading to an increase in the separation distance between the partials. Upon release of the external force, a reverse force occurs because the attractive force resulting from positive SF energy substantially outweighs the repulsive force due to the strain energy of the partials. This reverse force narrows the distance between the partials, diminishing gradually as the separation decreases (thereby reducing SF length). In a purely elastic scenario, the SF length will eventually recover to its initial state after unloading. In MPEAs, however, the scenario differs due to the presence of SRO, which may be disrupted by dislocation gliding – an inelastic process. The reduction in SRO by the leading partial alters the SF energy and therefore the reverse force upon unloading is also altered, leading to a rejuvenated state that provides an asymmetric loading and unloading with regards to the position of the leading partial. Upon retraction, there is an energetic balance between the partial dislocation restoring the SRO at the expense of then increasing the SFE. Further studies of a pure metal with a low SFE such as Cu would be interesting for comparison in this respect.

Fig. R3. Update the manuscript to address comment 1 of reviewer #3.